# Pretrained deep models outperform GBDTs in Learning-To-Rank under label scarcity

**Charlie Hou**[*]                                                                                     *charlieh@andrew.cmu.edu*
*Department of Electrical and Computer Engineering*
*Carnegie Mellon University*

**Kiran K. Thekumparampil**                                                                              *kkt@amazon.com*
*Amazon, Palo Alto*

**Michael Shavlovsky**                                                                                 *shavlov@amazon.com*
*Amazon, Palo Alto*

**Giulia Fanti**                                                                                       *gfanti@andrew.cmu.edu*
*Department of Electrical and Computer Engineering*
*Carnegie Mellon University*

**Yesh Dattatreya**                                                                                    *ydatta@amazon.com*
*Amazon, Palo Alto*

**Sujay Sanghavi**                                                                                  *sanghavi@mail.utexas.edu*
*Department of Computer Science*
*University of Texas at Austin*

**Reviewed on OpenReview:** *https://openreview.net/forum?id=093Q9VxaWt*

## Abstract

On tabular data, a significant body of literature has shown that current deep learning (DL) models perform at best similarly to Gradient Boosted Decision Trees (GBDTs), while significantly underperforming them on outlier data Gorishniy et al. (2021); Rubachev et al. (2022); McElfresh et al. (2023). However, these works often study problem settings which may not fully capture the complexities of real-world scenarios. We identify a natural tabular data setting where DL models can outperform GBDTs: tabular Learning-to-Rank (LTR) under label scarcity. Tabular LTR applications, including search and recommendation, often have an abundance of unlabeled data, and *scarce* labeled data. We show that DL rankers can utilize unsupervised pretraining to exploit this unlabeled data. In extensive experiments over both public and proprietary datasets, we show that pretrained DL rankers consistently outperform GBDT rankers on ranking metrics—sometimes by as much as 38%—both overall and on outliers.

## 1 Introduction

The learning-to-rank (LTR) problem aims to train a model to rank a set of items according to their relevance or user preference (Liu, 2009). An LTR model is typically trained on a dataset of queries and associated *query groups* (i.e., a set of potentially relevant *documents* or *items* per query), as well as an associated (generally incomplete) ground truth ranking of the items in the query group. The model is trained to output the optimal ranking of documents or items in a query group, given a query. LTR is a core ML technology in many real world applications—most notably in search contexts including Bing web search (Qin & Liu, 2013), Amazon

---

[*]Work partially done while Charlie Hou was an intern at Amazon.

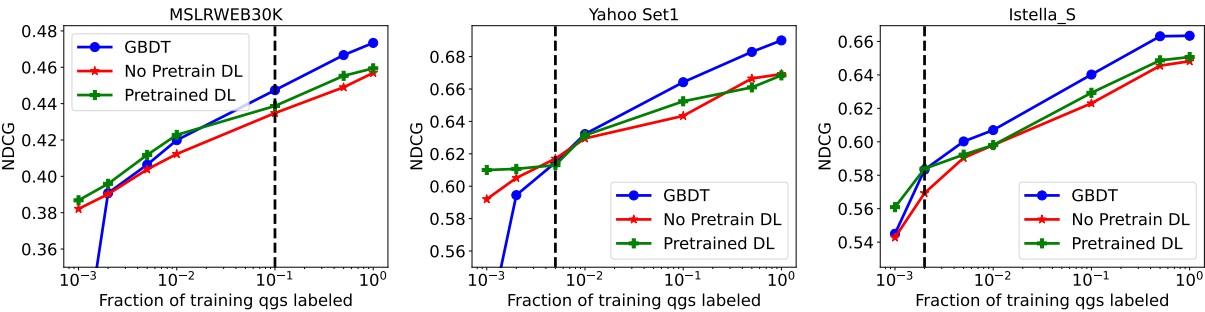

Figure 1: **Main result (Simulated crowdsourcing of labels):** We compare NDCG ($\uparrow$) as we sweep the percentage of labeled training query groups (QGs); labeled QGs have relevance scores for every element of the QG. For small enough fractions of labeled query groups, pre-trained DL rankers (sometimes significantly) outperform both GBDTs (representing the best of supervised GBDTs and semi-supervised GBDTs) and non-pretrained DL rankers (representing the best of tabular ResNet (Gorishniy et al., 2021), DeepFM (Guo et al., 2017), and DCNv2 (Wang et al., 2021)). To the left of the black dotted line, pretrained rankers perform the best (statistically significant at $p = 0.05$ level). Points are averages over 3 trials.

product search (Yang et al., 2022), and Netflix movie recommendations (Lamkhede & Kofler, 2021). More recently, LTR has also found use in Retrieval Augmented Generation (RAG), where a retrieval system is paired with an LLM (large language model) to ground the latter's responses to truth and reduce hallucination Lewis et al. (2020); Glass et al. (2022); Pan et al. (2022).

Traditionally, LTR models took *tabular features*—numerical or categorical features—of queries and documents as input features (Burges et al., 2005; Cao et al., 2007; Chapelle & Chang, 2011; Qin & Liu, 2013; Lucchese et al., 2016). As this continues to be the case for many important applications, *tabular LTR* is a core problem in machine learning (Bower et al., 2021; Yang et al., 2022; Yu, 2020; Kveton et al., 2022; Pan et al., 2022; Xia et al., 2008; Chen et al., 2009). Today, deep models are largely outperformed by gradient boosted decision trees (GBDTs) (Friedman, 2001) in ML problems with tabular features (Jeffares et al., 2023; Qin et al., 2021). In contrast, deep models are state-of-the-art by a significant margin in domains like text (Devlin et al., 2018) and images (He et al., 2016).

Recent breakthroughs in modeling non-tabular data like text and images have been driven by first training a deep neural network to learn from unlabeled data (unsupervised pretraining, or pretraining) (Devlin et al., 2018; Chen et al., 2020), followed by supervised training (finetuning). Models that are pretrained in this way can perform significantly better than models that were only trained on existing labeled data. This remarkable success stems from three factors: (1) limited access to labeled data, (2) large, available sources of unlabeled text and image data, and (3) pretraining methods that can exploit unlabeled data.

A natural question is whether deep models can similarly use unsupervised pretraining to outperform GBDTs on the tabular LTR problem. To this end, many papers have studied unsupervised pretraining techniques—both using deep learning and GBDTs—for tabular data Yoon et al. (2020); Ucar et al. (2021); Bahri et al. (2021); Verma et al. (2021); Majmundar et al. (2022); Duh & Kirchhoff (2008); Pseudo-Label (2013). However, none of these methods have shown convincingly that deep learning (with or without unsupervised pretraining) consistently outperforms GBDTs (semi-supervised or supervised); at best, deep methods appear to achieve roughly the same performance as GBDTs (Jeffares et al., 2023; Rubachev et al., 2022; Gorishniy et al., 2021; Shwartz-Ziv & Armon, 2022; Grinsztajn et al., 2022). Furthermore, recent work showed that GBDTs are significantly more robust against outliers in the dataset than deep learning models McElfresh et al. (2023). We note that most of these works study idealized problem settings which may fail to capture complexities of real-world scenarios such as label scarcity and noisy data. Prior work in other domains show that pretrained deep models can outperform purely supervised models Chen et al. (2020) and former models are more robust to noise Hendrycks et al. (2019). Therefore, we hypothesize that pretrained deep models can outperform GBDTs in tabular settings that more accurately reflect real-world challenges.

Through extensive empirical evaluation, we confirm this hypothesis and show that deep models which use unsupervised pretraining consistently outperform GBDTs—sometimes by a large margin—**when labels are scarce**. By scarce, we mean that only a small number of query groups contain labels. This setting is more standard in most real-world deployments of LTR, rather than the idealized settings studied in prior work. For instance, scarcity can arise when organizations: (a) crowdsource labels for a limited number of query groups due to cost constraints, and/or (b) use rare user feedback (e.g., clicks, purchases) as implicit labels. Our experiments consider both sources of scarcity. They are run on the three standard public datasets in the ranking literature: MSLRWEB30K (Qin & Liu, 2013), Yahoo (Chapelle & Chang, 2011), Istella_S (Lucchese et al., 2016), as well as an industry-scale proprietary ranking dataset.

**Contributions:** *(1) We demonstrate for the first time that unsupervised pretraining can produce deep models that outperform GBDTs in ranking.* Specifically, pretrained deep rankers can achieve up to 38% higher NDCG (normalized discounted cumulative gain) score (Burges, 2010) than GBDTs when labels are scarce (Figure 1). For instance, Figure 1 shows on three datasets that as the labels get scarcer, GBDTs start to substantially underperform pretrained deep rankers (experimental setup in Section 4.1.2).

*(2) We prescribe empirically-justified LTR-specific pretraining strategies*, including a new ranking-specific pretraining loss, SimCLR-Rank, which is a modification of the widely used SimCLR (Chen et al., 2020) loss. In experiments, SimCLR-Rank achieves similar or better performance than most existing pretraining methods at an order-of-magnitude speedup. This allows SimCLR-Rank to scale to large real-world datasets such as our proprietary industry-scale ranking dataset.

*(3) We demonstrate that when labels are scarce, pretrained deep rankers can perform significantly better than GBDTs on **outlier** query groups, boosting NDCG by as much as 80% (Figure 8*, Appendix A.3.4). This is perhaps surprising, given prior results in the label-rich regime showing that GBDTs are significantly more robust to data with irregularities than deep rankers (McElfresh et al., 2023).

The main novelty of this work lies in the identification of label-scarce LTR as a practical real-world tabular learning problem and the results which challenge the prior understanding of the optimal modeling choices for tabular learning. We demonstrate that pretraining strongly help deep models to consistently outperform GBDTs in this label-scarce problem. Of note, these improvements over GBDTs happen even when the labeled data is plentiful in absolute numbers; up to 25% labeled query groups in public datasets and 3 million+ labeled query groups in the large industry-scale ranking dataset. Prior to our work, the ML and ranking literature had not yet identified a widely applicable setting where (pretrained) deep models could consistently outperform GBDTs (Qin et al., 2021; McElfresh et al., 2023). Due to the prevalence of label scarcity in practical settings, our results suggest a positive change in how practitioners may approach real-world tabular learning problems, especially in search and recommendation systems.

**Related work:** We focus on the traditional LTR setting where the features are all numeric (tabular data). In this setting, gradient boosted decision trees (GBDTs) (Friedman, 2001) have historically been the de-facto models, though there is currently great interest in applying deep neural networks to tabular data. Borisov et al. (2022) categorize existing techniques for using deep neural networks over tabular data into four types: (1) **Encoding-based methods** such as VIME (Yoon et al., 2020), SCARF (Bahri et al., 2021), IGTD (Zhu et al., 2021), and SuperTML (Sun et al., 2019); (2) **Novel hybrid architectures** such as DeepFM (Guo et al., 2017), xDeepFM (Lian et al., 2018), and many others Cheng et al. (2016); Frosst & Hinton (2017); Ke et al. (2018; 2019); Popov et al. (2019); Luo et al. (2020); Liu et al. (2020); Ivanov & Prokhorenkova (2021); Luo et al. (2021); (3) **Transformer-based architectures** including SAINT (Somepalli et al., 2021), TabNet (Arik & Pfister, 2021), TabTransformer (Huang et al., 2020), and ARM-Net Cai et al. (2021)); (4) **Regularized DNNs** (Shavitt & Segal, 2018; Kadra et al., 2021).

Within these categories, proposed architectures include factorization machines (Rendle, 2010), wide & deep architectures (Cheng et al., 2016), and similar architectures designed for recommendation systems (Guo et al., 2017; Wang et al., 2021; Lian et al., 2018; Naumov et al., 2019; Qu et al., 2016). While our paper is architecture agnostic, we perform comparisons of pretrained models against non-pretrained DeepFM Guo et al. (2017) and DCNv2 Wang et al. (2021) models to show that pretraining is needed to outperform GBDTs consistently. Self-supervised learning (SSL) or unsupervised pretraining has improved overall performance

and robustness to noise (Hendrycks et al., 2019) in settings where there is a significant source of unlabeled data like text (Devlin et al., 2018; Howard & Ruder, 2018; Kitaev et al., 2018; Liu et al., 2017; Song & Roth, 2015) and images (Chen et al., 2020; Chen & He, 2021; Grill et al., 2020; Ge et al., 2021; Xie et al., 2020).

Inspired by the success of SSL in images and text, many unsupervised pretraining tasks have been proposed for tabular data. We describe some of the most well-known methods. SubTab (Ucar et al., 2021) proposes to train an autoencoder to reconstruct a table using a subset of the columns. VIME-self (Yoon et al., 2020) is a similar method. SCARF and DACL+ (Bahri et al., 2021; Verma et al., 2021) propose augmentations for the SimCLR pretraining task (Chen et al., 2020) that work for the tabular setting. RegCLR (Wang et al., 2022a) is a method for extracting tables from images, which is not directly applicable to our setting.

A smaller class of approaches has studied regularized DNNs, finding that they require limited changes to perform well on tabular data Shavitt & Segal (2018); Kadra et al. (2021). However, the most competitive of these methods require extensive hyperparameter optimization over a 14-dimensional grid Kadra et al. (2021).

While the body of work studying neural network architectures for tabular data is extensive, deep models have yet to outperform GBDTs convincingly (Qin et al., 2021; Joachims, 2006; Ai et al., 2019; Bruch et al., 2019; Ai et al., 2018; Pang et al., 2020; McElfresh et al., 2023). In this work, we consider encoding-based methods, which alter the representations over which a deep model can learn to rank; we do not focus on introducing new algorithms in this work, but rather evaluate mostly existing methods applied to tabular data. We comprehensively evaluate seven representative DL methods: SubTab, SCARF, DACL+, VIME-self, SimCLR-Rank, SAINT, SimSiam (Chen & He, 2021), and self-supervised approaches for GBDTs in Section 3.1. Our results show that multiple pretrained deep rankers can consistently outperform GBDTs in the label-scarce tabular LTR setting, with SimCLR-Rank and SimSiam (Chen & He, 2021) generally performing the best. We provide a detailed supplemental discussion of related works in the Appendix A.1.

## 2 Learning-To-Rank problem and its metrics

In the LTR problem, samples are query groups (QGs) consisting of a query (e.g. shopping query) and $L$ potentially relevant items (e.g. products) for this query. Although an LTR dataset contains multiple QGs, for the sake of simplicity, the notation in this section only handles a single QG. The $i$-th relevant item in this QG is represented by a feature vector $\boldsymbol{x}_i \in \mathbb{R}^d$, which captures information about this query-item pair, subsuming any query related features. If and only if the QG is labeled, such as the ones used in supervised training or testing, we also have an associated scalar relevance label $y_i$, which could be a binary, ordinal, or real-valued measurement of the relevance of this item for this query (Qin et al., 2021). The objective is to learn a function that ranks these $L$ items such that the items with highest true relevance are ranked at the top.

Most LTR algorithms formulate the problem as learning a scoring function $f_\theta : \mathbb{R}^d \to \mathbb{R}$ that maps the feature vector associated with each item to a score, and then ranking the items by sorting the scores in descending order. To measure the quality of a ranking induced by our scoring function $f_\theta$ on a labeled test QG, a commonly-used (per-QG) metric is NDCG (normalized cumulative discounted gain) ($\uparrow$):

$$\mathrm{NDCG}(\pi_{f_\theta}, \{y_i\}_{i=1}^L) = \frac{\mathrm{DCG}(\pi_{f_\theta}, \{y_i\}_{i=1}^L)}{\mathrm{DCG}(\pi^*, \{y_i\}_{i=1}^L)} \in [0, 1] \,, \tag{1}$$

where $\pi_{f_\theta} : [L] \to [L]$ ($[L] \triangleq \{1, \ldots, L\}$) is a ranking of the $L$ elements induced by the scoring function $f_\theta$ on $\{\boldsymbol{x}_i\}_{i=1}^L$ (i.e. $\pi(i)$ is the rank of the $i$-th item), while $\pi^*$ is the ideal ranking induced by the true relevance labels $\{y_i\}_{i=1}^L$, and discounted cumulative gain (DCG) is defined as $\mathrm{DCG}(\pi, \{y_i\}_{i=1}^L) = \sum_{i=1}^L \frac{2^{y_i}-1}{\log_2(1+\pi(i))}$. Here, ($\uparrow$) indicates that larger metric values are preferred. Note that for summarizing this metric over the full test set, we simply take the average across all the QGs in the set. Typically, a truncated version of NDCG is used that only considers the top-$k$ ranked items, denoted as NDCG@$k$. In the rest of our paper, we will refer to NDCG@5 as NDCG; this will be our evaluation metric.

We consider a natural label-scarce LTR setting with large amounts of unlabeled data and a limited supply of labels for a subset of this data. The labels that do exist can arise from multiple sources, such as crowdsourced

workers paid to label QGs, or through implicit user feedback (e.g., clicks or purchases); we model both in our evaluation. Unless specified otherwise, we use all the query groups for unsupervised pretraining, but we use only the labeled query groups for supervised training.

### 2.1 Outlier-NDCG for outlier performance evaluation

Motivated by a recent observation that GBDTs outperform DL models in datasets with irregularities McElfresh et al. (2023), we also study their performance on outliers. In interactive ML systems like search, performing well on outlier queries is particularly valuable as it empowers users to search for more outlier queries, which in turn allows the modeler to collect more data and improve the model. To this end, we evaluate the outlier performance by computing the average NDCG on outlier queries groups and refer to it as *Outlier-NDCG* for brevity. In practice, some outlier queries may already be known, and the modeler can define the outlier datasets accordingly, like in our proprietary dataset. For example, since industry data pipelines often have missing data/features one could identify samples with missing features as outliers. When outliers are unknown, detecting them (especially in higher-dimensional data) is challenging. Hence, we create a heuristic algorithm by assuming outliers are rare values that are separated from the bulk of the data. Using it we systematically selected outlier query groups for the public datasets that we use. More details about the outliers are provided in Appendix A.2.2. Separately measuring the performance on the outliers is justified by our later observation that performance on the full dataset and the outliers can significantly differ (Tables 2 and 3).

## 3 Design: Unsupervised pretraining for LTR

Qualitatively, the ranking relevance of tabular records to a query is typically determined by a combination of fields. For example, if a person searches for a product (say a pair of sandals), we can think of the fields in the LTR problem as representing properties of the results, such as material, color, and price. Qualitatively, we expect that if a pair of records are the same except for small differences between fields (e.g., missing a field), then the two products should generally have similar rankings in the LTR problem. In this section, we discuss how existing methods fail to capture this qualitative intuition.

**Semi-supervised learning for GBDTs:** While to the best of our knowledge there are no methods to perform unsupervised pretraining in GBDTs, one can use semi-supervised methods like consistency regularization (Jeong & Shin, 2020; Sajjadi et al., 2016; Miyato et al., 2018; Oliver et al., 2018; Berthelot et al., 2019), pseudo-labeling/self-training (Rizve et al., 2021; Pseudo-Label, 2013; Shi et al., 2018; Yarowsky, 1995; McClosky et al., 2006), and PCA (Duh & Kirchhoff, 2008) to address limited size of labeled data. In consistency regularization, one increases the size of the training set by using data augmentations. In pseudo-labeling/self-training, a model trained on the available labeled data is used to label unlabeled data. Then, the final model is trained on the resulting "labeled" dataset. In PCA, one projects the features along directions of maximum variability in the union of labeled and unlabeled datasets. Since prior work has found that consistency regularization decreases the performance of GBDTs (Rubachev et al., 2022), we evaluate only pseudo-labeling and PCA. None of these methods explicitly ensures that semantically similar records have similar representations in the learned models.

**Pretraining methods for tabular data:** To address this gap, there have been many unsupervised pretraining methods proposed for tabular data, which can be applied to the tabular LTR setting. In this paper, we will evaluate (1) SCARF (Bahri et al., 2021) and (2) DACL+ (Verma et al., 2021) which are contrastive learning methods (Chen et al., 2020), (3) VIME-self (Yoon et al., 2020) and (4) SubTab (Ucar et al., 2021) which are autoencoder based methods. The four most commonly-evaluated baselines in the pretraining literature for tabular data are Ucar et al. (2021); Hajiramezanali et al. (2022); Levin et al. (2022); Majmundar et al. (2022). Our results suggest that many of these methods outperform GBDTs in the label-scarce regime (Section 3.1), in accordance with the main theme of this paper. However, the consistently best-performing methods in our experiments were more basic, domain-agnostic pretraining methods, which can be viewed as more primitive building blocks of some of the above methods. Surprisingly, they use *less* domain knowledge of tabular data than these more sophisticated methods.

Specifically, in the image setting, SimCLR (Chen et al., 2020) and SimSiam (Chen & He, 2021) are two widely-adopted unsupervised pretraining methods. They use the idea that two variations of the same data point should have similar representations. This idea generalizes across domains, including tabular LTR.

*SimCLR (Chen et al., 2020):* In a batch of $B$ query-groups, the feature vector $\boldsymbol{x}_{q,i}$ of $q$-th query group and its $i$-th item is stochastically augmented twice to obtain $\boldsymbol{x}_{q,i}^{(0)}$ and $\boldsymbol{x}_{q,i}^{(1)}$, which are called a *positive pair*. In this paper, we use combinations of the following two augmentation strategies for all domain-agnostic pretraining methods: *(a) Zeroing:* Zero out features selected randomly and i.i.d. with fixed probability. *(b) Gaussian noise:* Add Gaussian noise of a fixed scale to all features. Further details are given in Appendix A.2.3.

Next, a base encoder $h(\cdot)$ and projection head $g(\cdot)$ map $\boldsymbol{x}_{q,i}^{(a)}$ to $\boldsymbol{z}_{q,i}^{(a)} = g(h(\boldsymbol{x}_{q,i}^{(a)}))$ for $a = 0, 1$. Then by minimizing the InfoNCE loss (Oord et al., 2018) in this projected representation space, we push the positive pair, $(\boldsymbol{z}_{q,i}^{(0)}, \boldsymbol{z}_{q,i}^{(1)})$, closer while pushing them farther from all other augmented data points in the batch, which are called their *negatives*. Formally, for $\boldsymbol{x}_{q,i}$, we minimize $\ell_{q,i}^{(0)} + \ell_{q,i}^{(1)}$, where:

$$\ell_{q,i}^{(a)} = -\cos(\boldsymbol{z}_{q,i}^{(a)}, \boldsymbol{z}_{q,i}^{(\bar{a})})/\tau + \log \sum_{q'=1}^{B} \sum_{i'=1}^{L_{q'}} \sum_{a'=0}^{1} \left[ \mathbf{1}\{(q', i', a') \neq (q, i, a)\} \cdot \exp(\cos(\boldsymbol{z}_{q,i}^{(a)}, \boldsymbol{z}_{q',i'}^{(a')})/\tau) \right] \quad (2)$$

$\bar{a} = 1 - a$, $\tau$ is the temperature parameter, $L_{q'}$ is the number of items in $q'$-th query group, and $\cos(\boldsymbol{z}, \widetilde{\boldsymbol{z}}) = \langle \boldsymbol{z}, \widetilde{\boldsymbol{z}} \rangle / \|\boldsymbol{z}\|\|\widetilde{\boldsymbol{z}}\|$ denotes the cosine similarity. After pretraining, the encoder $h$ is used in downstream applications. SimCLR achieves superior performance in many domains (Chen & He, 2021; Wang et al., 2022b;a). It is known that contrasting a pair of data samples which are hard to distinguish from each other, called *hard negatives* can help an encoder learn good representations (Robinson et al., 2020; Oh Song et al., 2016; Schroff et al., 2015; Harwood et al., 2017; Wu et al., 2017; Ge, 2018; Suh et al., 2019). SimCLR simply contrasts all pairs of data samples in the batch (including from other query groups) with the assumption that a large enough batch is likely to contain a hard negative.

*SimSiam (Chen & He, 2021):* Similar to SimCLR, for each data point $\boldsymbol{x}_{q,i}$, this method produces stochastically-augmented *positive pairs* and their *projected* representations. However, we pass these representations further through a predictor pred($\cdot$), to get $\boldsymbol{p}_{q,i}^{(a)} = \mathrm{pred}(\boldsymbol{z}_{q,i}^{(a)})$ for $a = 0, 1$. Finally, we maximize the similarity between the projected and predicted representations: $\cos(\boldsymbol{p}_{q,i}^{(0)}, \mathrm{sg}(\boldsymbol{z}_{q,i}^{(1)})) + \cos(\boldsymbol{p}_{q,i}^{(1)}, \mathrm{sg}(\boldsymbol{z}_{q,i}^{(0)}))$, where *sg* is a stop-gradient. Unlike SimCLR, there are no *negatives*, i.e., the loss function does not push the representation of an augmented sample away from that of other samples' augmentations. The asymmetry in the loss due to the stop gradient and the predictor prevents representation collapse to make negatives unnecessary Tian et al. (2021); Zhang et al. (2021); Wang et al. (2022b). If the number of items in each query group is $L$, then the time/space complexity for SimSiam is only $O(BL)$ per batch, whereas it is $O(B^2 L^2)$ for SimCLR. Therefore, SimSiam is more efficient and it can scale to larger batchsizes and data.

**Pretraining method for LTR:** Motivated by (a) SimCLR's high complexity and (b) the efficacy of contrasting with hard negatives, we additionally propose SimCLR-Rank, an LTR-specific alternative. Recall that the SimCLR loss for an item in equation 2 uses all the other items in the batch as negatives. Good pretraining strategies usually exploit the structure of the data, e.g. text sequences motivates masked token prediction Devlin et al. (2018). Studying the query group structure of LTR data, we notice that high-quality hard negatives are freely available as *the other items from the same query group*. These items are retrieved and deemed potentially similarly relevant for this query by an upstream retrieval model, making these items harder negatives than other items in the batch. So, we propose SimCLR-Rank, which modifies SimCLR to contrast only with the other items in the same query group, as illustrated in Figure 2. Formally, we modify the per-augmentation loss $\ell_{q,i}^{(a)}$ as:

$$\ell_{q,i}^{(a)} = -\cos(\boldsymbol{z}_{q,i}^{(a)}, \boldsymbol{z}_{q,i}^{(\bar{a})})/\tau + \log \sum_{i'=1}^{L_q} \sum_{a'=0}^{1} \left[ \mathbf{1}\{(i', a') \neq (i, a)\} \cdot \exp(\cos(\boldsymbol{z}_{q,i}^{(a)}, \boldsymbol{z}_{q,i'}^{(a')})/\tau) \right] \quad (3)$$

Empirically, SimCLR-Rank better separates embeddings within query groups relative to sampling negatives uniformly from each batch (Appendix A.5). The SimCLR-Rank loss has a time/space complexity of $O(BL^2)$,

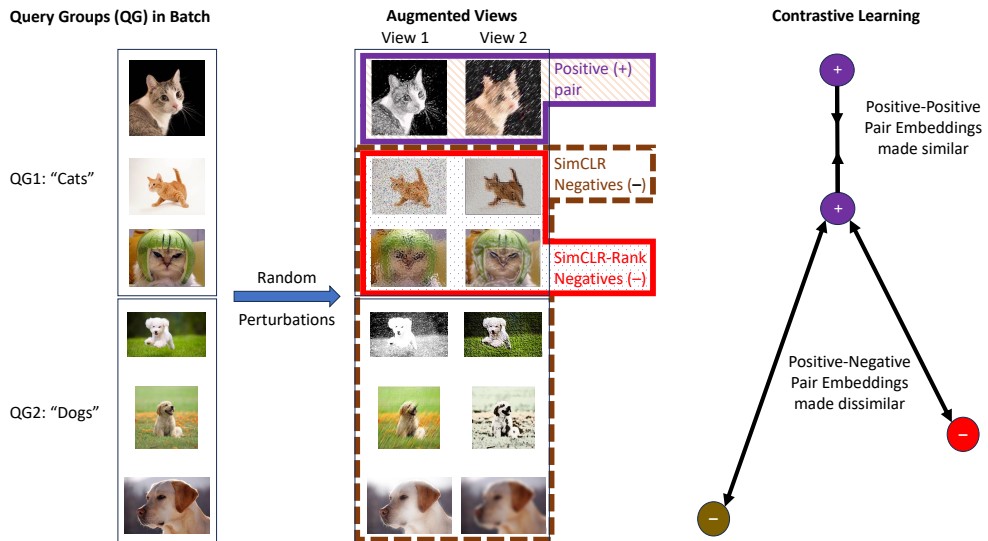

Figure 2: In SimCLR and its variants, *positive pairs*, or augmented versions of the same data sample, are trained to have similar embeddings. Positive-negative pairs, or augmented samples originating from two different data points or classes, are trained to have distant embeddings. In vanilla SimCLR, each positive pair is contrasted with *all other items in the batch*, denoted by the data points contained in the brown dashed line above. In SimCLR-Rank, each positive pair is contrasted with *only the items in the same Query Group (QG)*, denoted by the data points inside the red solid line above.

Table 1: **Pretraining methods:** A comparison across unsupervised pretraining methods on NDCG, averaged over 3 trials. Here only 0.1% of the query groups have labels for finetuning. **Bold** numbers denote the best in a column, and underlined numbers are within the margin of error of the best. Given that SCARF/DACL+/SimCLR are too slow to scale to large LTR datasets (for example on the industrial-scale proprietary online shopping dataset), we find that SimCLR-Rank and SimSiam are the best pretraining methods for tabular LTR.

| Method | MSLR ($\uparrow$) | Yahoo Set1 ($\uparrow$) | Istella ($\uparrow$) |
|---|---|---|---|
| Supervised GBDT | $0.2801 \pm 0.0002$ | $0.5083 \pm 0.0197$ | $0.5450 \pm 0.0000$ |
| Semi-supervised GBDT | $0.2839 \pm 0.0004$ | $0.5061 \pm 0.0267$ | $0.4656 \pm 0.0310$ |
| SimCLR-Rank + GBDT | $0.3165 \pm 0.0050$ | $0.5504 \pm 0.0135$ | $0.5397 \pm 0.0098$ |
| SimSiam + GBDT | $0.3158 \pm 0.0030$ | $0.5620 \pm 0.0167$ | $0.5297 \pm 0.0051$ |
| SCARF (Bahri et al., 2021) | $0.3807 \pm 0.0016$ | $0.5884 \pm 0.0129$ | $0.5542 \pm 0.0024$ |
| DACL+ (Verma et al., 2021) | $\underline{0.3833 \pm 0.0037}$ | $0.5887 \pm 0.0012$ | $\mathbf{0.5626 \pm 0.0024}$ |
| VIME-self (Yoon et al., 2020) | $0.3834 \pm 0.0011$ | $0.5839 \pm 0.0058$ | $0.5514 \pm 0.0025$ |
| SubTab (Ucar et al., 2021) | $0.3748 \pm 0.0025$ | $0.5814 \pm 0.0062$ | $0.5082 \pm 0.0050$ |
| SAINT (Somepalli et al., 2021) | $0.3355 \pm 0.0043$ | $0.5890 \pm 0.0075$ | $0.5560 \pm 0.0066$ |
| SimCLR (Chen et al., 2020) | $0.3827 \pm 0.0027$ | $0.5837 \pm 0.0093$ | $\underline{0.5602 \pm 0.0055}$ |
| SimCLR-Sample | $0.3498 \pm 0.0541$ | $0.5803 \pm 0.0070$ | $0.5479 \pm 0.0016$ |
| SimCLR-Rank | $\mathbf{0.3868 \pm 0.0026}$ | $0.5843 \pm 0.0062$ | $\underline{0.5609 \pm 0.0040}$ |
| SimSiam (Chen & He, 2021) | $0.3790 \pm 0.0028$ | $\mathbf{0.6100 \pm 0.0072}$ | $0.5189 \pm 0.0096$ |

making it faster than SimCLR's $O(B^2L^2)$ complexity, and enjoying SimSiam's linear complexity in $B$. Next, we empirically study these methods.

## 3.1 Evaluation of unsupervised pretraining techniques

In this subsection, we compare the discussed pretraining strategies in the following tabular LTR setting.

**Dataset.** We use MSLRWEB30K (or MSLR for brevity) (Qin & Liu, 2013), Yahoo Set1 (or Set1 for brevity) (Chapelle & Chang, 2011), and Istella_S (Lucchese et al., 2016) (a smaller version of Istella (Dato et al., 2016)). In the tabular LTR literature, these three datasets are generally considered the standard; most recent papers evaluate (only) on these datasets (Qin et al., 2021; Ai et al., 2019; Pang et al., 2020; Yang et al., 2022; Yu, 2020). To simulate scarcity of labels, we let 0.1% of the training query groups in each dataset keep their labels, while the rest of the query groups have no labels. Dataset statistics are provided in Appendix A.2.1.

**Methodology.** For the neural pretraining approaches (which include SimCLR-Rank, SimSiam, SimCLR, SubTab, VIME-self, DACL+, SAINT, and SCARF), the base of the model is the tabular ResNet[1] (check the footnote for our explanation on the architecture choice). We also evaluate "SimCLR-Sample", a neural pretraining baseline we create to demonstrate the value of negatives from the same query group. SimCLR-Sample is like SimCLR-Rank, but at the beginning of each epoch the items are first randomly permuted and grouped into fake query groups. Then we apply SimCLR-Rank. Thus SimCLR-Sample has the same time/space complexity as SimCLR-Rank. We pretrain on all the available query groups and finetune/train on only the labeled query groups.

For the GBDT-based approaches, we use the implementation in `lightgbm` Ke et al. (2017) and adopt a tuning strategy similar to Qin et al. (2021). We report "Supervised GBDT" as GBDT utilizing only the labeled data, and in "Semi-supervised GBDT" we report the best among pseudo-labeled (Pseudo-Label, 2013) and PCA-enriched (Duh & Kirchhoff, 2008) GBDTs as well as the combination of the two techniques. We also evaluate GBDTs enriched with SimCLR-Rank and SimSiam embeddings as additional features, which we call "SimCLR-Rank + GBDT" and "SimSiam + GBDT". We describe our experimental setup in detail in Appendix A.2.3. This includes hyperparameters, procedures for data augmentation, GBDT pseudo-labeling/PCA details, model training, and finetuning. For comparison to prior baselines on tabular data, our experimental setup is chosen to mimic the original setup in them.

**Results.** Table 1 charts the test NDCG for each of the pretraining methods we study. We highlight a few observations: (1) The pretraining-based DL methods almost all outperform GBDTs (both supervised and semi-supervised) in this label-scarce setting. Surprisingly, the semi-supervised GBDT baselines tend to achieve lower NDCG than the supervised GBDT baseline without semi-supervised learning. (2) When GBDTs are allowed to use representations learned from our pretraining methods, their performance consistently improves, though they do not outperform pretraining with DL. (3) SimCLR-Rank and SimSiam perform the best or the second best among these methods for most of the datasets. In particular, SimCLR-Rank outperforms SimCLR-Sample, demonstrating the value and efficiency of negatives from the same query group. As expected, SimCLR-Rank is also 7-14x faster than SimCLR/SCARF/DACL+ (Table 6 in Appendix A.3.1). Given these advantages in terms of test NDCG and speed, we select SimCLR-Rank and SimSiam as the best overall pretraining methods for tabular LTR, and use them for our continued evaluation.

## 4 Empirical evaluation

Next we compare the best DL pretraining methods for tabular LTR (SimCLR-Rank and SimSiam) against GBDT and non-pretrained DL models on (1) the public datasets under different label scarcity patterns to simulate real-world scenarios[2], and (2) a private large-scale ranking dataset which is naturally label-scarce.

### 4.1 Public datasets results

For the public datasets (MSLRWEB30K, Yahoo Set1, Istella_S; details given in Section 3.1), we simulate label scarcity occurring in two scenarios: (1) *crowdsourcing* of labels for a limited number of query groups, and (2) using scarce *implicit binary user feedback* (e.g. clicks) as labels.

---

[1]We emphasize that we do not prescribe any specific deep learning architecture choice; the choice for the best DL model may depend on the specific LTR task. Our proposed workflow of pretraining deep models in LTR is agnostic to the specific choice of DL architecture. Prior work has proposed DL architectures designed for click-through rate (CTR)/LTR based on factorization machines (Rendle, 2010), such as DeepFM and DCNv2 (Guo et al., 2017; Wang et al., 2021), which we compare to in Section 4.1.2 and Section 4.1.2 against our pretrained models. Note that it is unclear currently how to do unsupervised pretraining using these CTR/LTR specific architectures at the moment.

[2]Code is provided at `https://github.com/houcharlie/ltr-pretrain/`.

### 4.1.1 Simulated crowdsourcing of labels

We simulate a setting where labels are crowdsourced for only some query groups due to limited resources.

**Dataset.** For each of the three public datasets, we vary the fraction of labeled query groups in the training set in $\{0.001, 0.002, 0.005, 0.1, 0.5, 1.0\}$. Note that within each labeled query group all items are labeled.

**Methodology.** We repeat the methodology from Section 3.1 for the GBDT and the SimCLR-Rank and SimSiam pretraining methods. For each dataset and labeling fraction, we report the test metric of the best GBDT ranker among semi-supervised GBDTs and supervised GBDTs under "GBDT". Similarly, under "no pretrain DL" we report the best supervised DL model amongst tabular ResNet (Gorishniy et al., 2021), DeepFM (Guo et al., 2017), and DCNv2 (Wang et al., 2021) under the name "no pretrain DL". Finally, we report the best pretrained ranker amongst SimCLR-Rank and SimSiam under the name "pretrained DL". Pretrained DL rankers use tabular ResNet (see Appendix A.2.3 for more details). During hyperparameter tuning, we tune for NDCG and report the resulting test NDCG and test Outlier-NDCG.

**Results.** We compare pretrained rankers, non-pretrained DL rankers, and GBDTs on public datasets in Figure 1. The results for Outlier-NDCG are provided in Figure 8 in Appendix A.3.4. We find that pretrained rankers outperform non-pretrained methods (including GBDTs) on NDCG and Outlier-NDCG across all public datasets, up to a dataset-dependent fraction of QGs labeled, as shown in Figure 1 (for NDCG) and Figure 8 (for Outlier-NDCG). We also note that GBDTs have better NDCG than non-pretrained DL models in most of these regimes. This shows that there exist scenarios with a limited supply of labeled query groups, where self-supervised pretraining is the factor that allows deep learning models to outperform GBDTs. We provide a detailed comparison between semi-supervised and supervised GBDT models in Appendix A.4.

### 4.1.2 Simulated implicit binary user feedback

Here we simulate a scenario when users of an LTR system provide binary implicit feedback for the items, like the clicking of an advertisement on a webpage. This kind of feedback is usually infrequent in an LTR system, as it is the result of a user choosing to spend more time or resources on an item in a presented list. Therefore, most query groups will not have any labels, and even labeled query groups will only have a few positively labeled items. On the other hand, such labels are often cheaper to collect than crowdsourced labels, so they are used in many industry use cases.

**Dataset.** To simulate this scenario, we follow the methodology from Yang et al. (2022), to generate independent stochastic binary labels for each item from its true relevance label for training and validation sets of each of the public datasets. Note that we still use the true (latent) relevance labels in the test set for evaluation. This models a scenario where we observe binary labels that are noisy observations of a true latent label, but the task is to use these noisy labels to learn to rank according to the true labels. The details of this binarizing transformation are provided in Appendix A.2.4. In this transformation, a parameter, $\tau_{\text{target}}$, implicitly controls how sparse the binary labels are (larger is more sparse). We select $\tau_{\text{target}} = 4.5$, to obtain 8.9%, 5.3%, and 25.2% labeled query groups which contain at least one positive label in MSLR, Yahoo Set1, Istella_S datasets, respectively. We present detailed data statistics in Table 11 of Appendix.

**Methodology.** We reuse the methodology of Section 4.1.1, except we use a linear scoring head for pretrained models to improve stability and performance (see Appendix A.2.3). Similar to Section 4.1.1, we report the best of multiple models for each method and dataset (see Section 4.1.1 for details).

**Results.** In Table 2, we present the test NDCG and Outlier-NDCG metrics of the models under this setting. We see that pretrained rankers significantly outperform GBDTs and non-pretrained DL rankers in this simulated binary user feedback setting with up to, even with 25.2% labeled query groups (Istella_S). For MSLRWEB30K and Yahoo Set1 datasets, the second best model is the GBDT, and the non-pretrained DL ranker has a much worse Outlier-NDCG. This aligns with recent observations that GBDTs perform better than DL models in supervised training with datasets containing irregularities McElfresh et al. (2023). However, pretraining helps DL rankers close this gap, and at times beat GBDTs in outlier performance. We also present results for $\tau_{\text{target}} \in \{4.25, 5.1\}$ in Appendix A.3.5. There, we find that making the labels sparser ($\tau_{\text{target}} = 5.1$) increases the relative improvement of pretrained DL rankers, while in a less sparse

Table 2: **Main result (Simulated implicit binary user feedback):** When $X\%$ (MSLRWEB30K=8.9%, Yahoo Set1=5.3%, and in Istella_S=25.2%) of query groups are assigned a few binary labels, pretrained DL method outperform both GBDT and non-pretrained DL models. We generate labels with $\tau_{\text{target}} = 4.5$ (Section 4.1.2). ♣ means that pretrained models are significantly better in NDCG than non-pretrained models, measured by a t-test with significance $p < 0.05$.

| Method | MSLRWEB30K (↑) | Yahoo Set1 (↑) | Istella (↑) |
|---|---|---|---|
| | NDCG (↑) | | |
| GBDT | $0.346 \pm 0.001$ | $0.617 \pm 0.000$ | $0.602 \pm 0.000$ |
| No Pretrain DL | $0.333 \pm 0.001$ | $0.621 \pm 0.000$ | $0.608 \pm 0.002$ |
| Pretrained DL | $\mathbf{0.356 \pm 0.004}$♣ | $\mathbf{0.622 \pm 0.001}$ | $\mathbf{0.613 \pm 0.002}$♣ |
| | Outlier-NDCG (↑) | | |
| GBDT | $0.288 \pm 0.000$ | $0.540 \pm 0.000$ | $0.678 \pm 0.000$ |
| No Pretrain DL | $0.238 \pm 0.026$ | $0.536 \pm 0.003$ | $0.687 \pm 0.015$ |
| Pretrained DL | $\mathbf{0.289 \pm 0.012}$ | $\mathbf{0.543 \pm 0.002}$♣ | $\mathbf{0.699 \pm 0.003}$ |

Table 3: **Main result (Real-world dataset):** % improvement of rankers pretrained by SimSiam and SimCLR-Rank over GBDTs on a large-scale proprietary online shopping dataset. Unsupervised pretraining improves performance (1) overall (full dataset) (internally, 2% is a big improvement), and (2) on outliers.

| Method | NDCG (↑) | Outlier-NDCG (↑) |
|---|---|---|
| | $\Delta\%$ change w.r.t. GBDT | |
| GBDT | +0.00% | +0.00% |
| No Pretrain DL | $+3.82\% \pm 0.14\%$ | $-6.97\% \pm 0.43\%$ |
| SimCLR-Rank | $+4.00\% \pm 0.14\%$ | $-4.85\% \pm 0.44\%$ |
| SimSiam | $\mathbf{+5.64\% \pm 0.14\%}$ | $\mathbf{+2.75\% \pm 0.43\%}$ |

setting ($\tau_{\text{target}} = 4.25$) GBDTs dominate. This suggests that pretrained DL models outperform GBDTs when learning from rare implicit binary user feedback, as is common in search and recommendation systems.

## 4.2 Large-scale real-world dataset results

In this section we test whether the results from the simulated label scarce settings above, especially Section 4.1.2, translate to a large-scale real-world LTR problem.

**Dataset.** Our proprietary dataset is derived from the online shopping logs of a large online retailer, numbering in the tens of millions of query groups. In this dataset, query groups consist of items presented to a shopper if they enter a search query. We assign each item a purchase label: label is 1 if the shopper purchased the item, or 0 if the shopper did not purchase the item. Hence, this is a real-world instance of the implicit binary user feedback setting simulated in Section 4.1.2. Further, only about 10% of the query groups have items with non-zero labels. See Appendix A.2.1 for more dataset details.

**Methodology.** We compare the performance of pretrained DL models–SimCLR-Rank and SimSiam–with GBDT and non-pretrained DL models. SimSiam and SimCLR-Rank are first pretrained on all the query groups before being finetuned on the labeled query groups with some non-zero purchase labels. The GBDT and the non-pretrained models are only trained on the labeled data. More details about these models and training strategies is given in Appendix A.2.5.

**Results.** Our results are summarized in Table 3 and all numbers are given as relative percentage improvements ($\Delta\%$) over the GBDT ranker. First, note that all the DL models outperform GBDT models in terms of NDCG. Amongst them, the pretrained model using SimSiam performs the best with a gain of over 5.5%. This is substantial, given that in large-scale industry datasets, even a 2% improvement in NDCG is considered significant. In terms of Outlier-NDCG, the story is very different. Here, the non-pretrained DL model and

SimCLR-Rank performs up to -7% worse than the GBDT. However, SimSiam pretraining allows the DL ranker to overcome this shortcoming and beat the outlier performance of the GBDT. This verifies our hypothesis from Section 4.1.2 that pretraining can help DL models outperform GBDTs on large-scale real-world LTR problems, especially with implicit user feedback. Lastly, we see a significant performance difference between SimSiam and SimCLR-Rank. We investigate this in an ablation with public datasets in Section 4.3.

### 4.3 Ablations: Choices for pretrained models in LTR

Our experiments suggest that the best pretraining and finetuning strategies for the LTR problem are different from the known best practices for image or text tasks. We make a few observations. (1) SimSiam and SimCLR-Rank perform significantly differently depending on the dataset. (2) Combining models pretrained by SimCLR-Rank and SimSiam can produce a model that is competitive against SimCLR-Rank and SimSiam across all datasets, producing a unified approach that may be performant across many settings. (3) Full finetuning of the model performs much better than *linear probing*, which only finetunes a linear layer over frozen pretrained embeddings. In the remainder of this section, we elaborate on these findings in more detail.

**(1) SimCLR-Rank and SimSiam perform substantially differently depending on the dataset.** SimSiam and SimCLR-Rank perform significantly differently depending on the dataset, which is different from the observations of Chen & He (2021), who proposed SimSiam and found that it performs similarly to SimCLR derivatives. We next compare SimCLR-Rank with SimSiam at different levels of label scarcity.

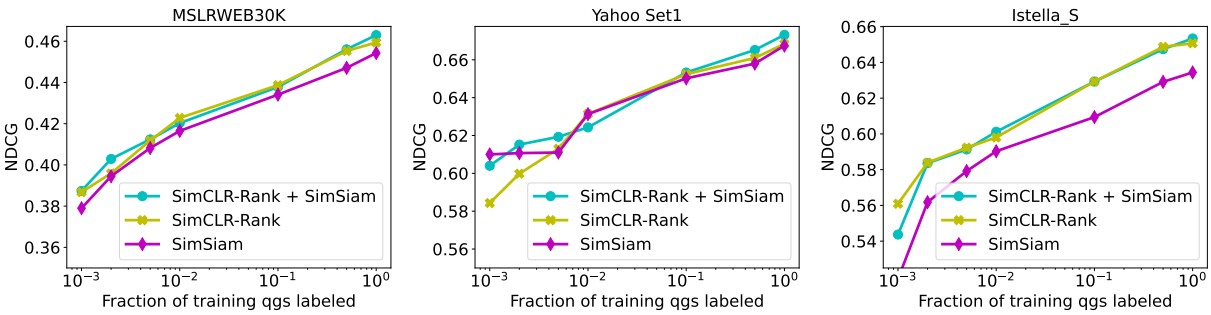

Figure 3: Comparison of SimCLR-Rank and SimSiam pretraining strategies on three public datasets with varying fraction of labeled query groups. Neither method consistently dominate the other in all the datasets. However, by combining SimCLR-Rank and SimSiam encoders, we can produce models that are competitive across all datasets. Data points are averages over 3 trials.

*Methodology.* The dataset and methodology follows that of Section 4.1.1, except we let (1) SimCLR-Rank use the Gaussian augmentation with scale 1, and (2) SimSiam use the zeroing augmentation with probability 0.1. These augmentations generally perform well for their respective methods. Experiments illustrating the effect of each augmentation method are included in Appendix A.3.6.

*Results.* In Figure 3, SimCLR-Rank performs better on MSLRWEB30K and Istella_S, while SimSiam performs better on Yahoo Set1 when labels are scarce. Figure 4 shows the t-SNE projections of sub-sampled item embeddings learned by SimCLR-Rank, SimSiam, and pre-final layer of a fully supervised model for Yahoo Set1. SimSiam exhibit qualitatively more clustered embeddings than SimCLR-Rank, particularly for high relevance labels. Similar t-SNE plots for other datasets are in Appendix A.3.3.

**(2) Combining models pretrained by SimCLR-Rank and SimSiam produces a model that has the strengths of both.** To make steps towards a single recommendation for unsupervised pretraining in LTR, we unify SimCLR-Rank and SimSiam into a single method "SimCLR-Rank + SimSiam".

*Methodology.* To produce a "SimCLR-Rank + SimSiam" model, we finetune a linear layer over the embeddings from a finetuned SimCLR-Rank concatenated with the embeddings from a finetuned SimCLR-Rank model. More details about how we combine the models is given in Appendix A.3.7.

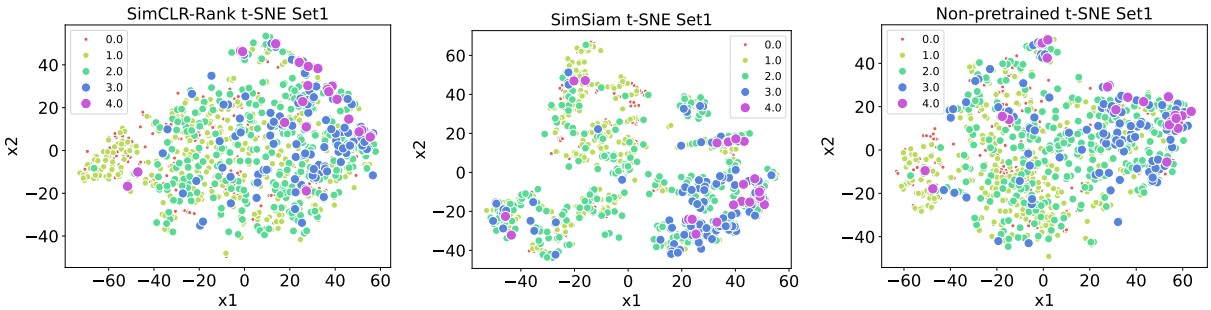

Figure 4: t-SNE plots of SimCLR-Rank, SimSiam, and a non-pretrained model trained on 0.1% of the data for the Yahoo Set1 dataset. Both SimCLR-Rank and SimSiam use the zeros augmentation ($p = 0.1$) to remove augmentation choice as a confounding variable. Relevance label is indicated by marker size/color. We see that SimCLR-Rank and SimSiam embeddings are qualitatively very different. Qualitatively, SimCLR-Rank and SimSiam embeddings separate different relevance labels better than the non-pretrained model. For this setting, SimSiam has the best downstream performance, and also gives the most clustered embeddings.

*Results.* In Figure 3 we find that "SimCLR-Rank + SimSiam" performs equal to or better than SimCLR-Rank and SimSiam across many different percentages of labeled query groups. While the combined model does not uniformly dominate across all labeled query group percentages, it is a promising first step towards a single unified method for unsupervised pretraining in ranking.

**(3) Full finetuning outperforms linear probing.** Linear probing is a popular finetuning strategy in text and image domains, where it produces good results (Chen & He, 2021; Chen et al., 2020; Peters et al., 2019; Kumar et al., 2022). We next compare it with the full finetuning strategy in tabular LTR.

*Methodology.* We use the methodology from Section 3.1 to compare 3 finetuning strategies: linear probing (LP), full-finetuning (FF), and multilayer probing (MP) for SimSiam and SimCLR-Rank. MP tunes a 3-layer MLP head on top of the frozen pretrained embeddings. Experimental details are in Appendix A.2.3.

*Results.* Table 4 provides results on MSLRWEB30K dataset. We find that in terms of NDCG of both SimCLR-Rank and SimSiam, LP is the worst and FF is the best. Interestingly, MP comes as a close second in SimCLR-Rank, but it performs poorly on SimSiam. Our results on other datasets are similar (Appendix A.3.2). We thus recommend full fine-tuning as a stable and performant strategy. To explain the above observations, we we can look again at the t-SNE projections of the embeddings generated by the SimSiam and SimCLR-Rank encoders (pretrained using the zeros augmentation with $p = 0.1$ to remove augmentation choice as a confounding factor, finetuned on 0.1% of the data) and the pre-final layer of a model trained only on 0.1% of the data on a randomly sampled 1000 samples from the full training set (Figure 4).

These plots offer the following explanations for the two phenomena. (1) The embeddings produced by the SimSiam/SimCLR-Rank are not sorted by relevance score in the projection space, unlike the fully supervised encoder. This suggests that a linear ranker cannot use the pretrained embeddings to predict the true relevance labels. (2) MP works better for SimCLR-Rank than SimSiam because SimCLR-Rank's embeddings are more evenly spread out (less collapsed) than those of SimSiam. This allows an MLP trained on the SimCLR-Rank embeddings to distinguish the items with different relevance labels.

## 5 Conclusion

We study the learning-to-rank (LTR) problem with tabular data under a scarcity of labeled data—a common scenario in real-world practical LTR systems. Prior works on supervised learning with tabular data have shown that GBDTs outperform deep learning (DL) models, especially on datasets with outliers. In this paper, we find that in the label-scarce setting, DL pretraining methods can exploit available unlabeled data to obtain new state-of-the art performance. Through experiments with public LTR datasets and a real-world large-scale

Table 4: **Finetuning**: Comparison of fine tuning strategies on MSLRWEB30K dataset (LP = Linear Probing, MP = Multilayer Probing, FF = Full Finetuning). Full finetuning consistently performs best or near-best, while linear probing *and* multilayer probing perform poorly.

| Method | SimSiam | SimCLR-Rank |
|--------|---------|-------------|
| *NDCG* ($\uparrow$) | | |
| LP | $0.2679 \pm 0.0007$ | $0.3219 \pm 0.0224$ |
| MP | $0.2764 \pm 0.0001$ | $0.3890 \pm 0.0011$ |
| FF | $\mathbf{0.3935 \pm 0.0034}$ | $\mathbf{0.3959 \pm 0.0022}$ |
| *Outlier-NDCG* ($\uparrow$) | | |
| LP | $0.1803 \pm 0.0033$ | $0.2304 \pm 0.0332$ |
| MP | $0.1749 \pm 0.0023$ | $\mathbf{0.2969 \pm 0.0009}$ |
| FF | $\mathbf{0.3149 \pm 0.0119}$ | $0.2892 \pm 0.0025$ |

online shopping dataset, we show that pretraining allows DL rankers to outperform GBDT rankers, especially on outlier data. Finally, we provide guidelines for pretraining on LTR datasets. There remain several open questions around pretraining for tabular and LTR data. It is still unknown whether we can achieve knowledge transfer across different tabular datasets, as is commonly done in image and text domains. It will also be useful to investigate LTR-specific pretraining methods which can further improve the ranking performance.

## Acknowledgments

GF and CH gratefully acknowledge the support of Google, Intel, the Sloan Foundation, and Bosch.

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

# A Appendix

## A.1 Detailed discussion of related work

**Learning-To-Rank.** In our paper, we focus on the traditional LTR setting where the features are all numeric (tabular data). However, there is a line of work in LTR where raw text is also an input. In this case, one can leverage large language models in the ranking setting (Zhuang et al., 2023; Han et al., 2020; Yates et al., 2021; Nogueira et al., 2019; Mitra et al., 2018).

In tabular LTR problems, the dominant models currently used are gradient boosted decision trees (GBDTs) (Friedman, 2001), which are not deep learning models. GBDT models, which perform well on tabular data,

are adapted to the LTR setting via losses that are surrogates for ranking metrics like NDCG. Surrogate losses (including LambdaRank/RankNet (Burges, 2010) and PiRank (Swezey et al., 2021)) are needed because many important ranking metrics (like NDCG) are non-differentiable. The combination of tree-based learners and ranking losses has become the de-facto standard in ranking problems, and deep models have yet to outperform them convincingly (Qin et al., 2021; Joachims, 2006; Ai et al., 2019; Bruch et al., 2019; Ai et al., 2018; Pang et al., 2020).

**Deep tabular models.** Given the success of neural methods in many other domains, there have been many attempts to adapt deep models to the tabular domain.

Borisov et al. (2022) categorize existing techniques for using deep neural networks over tabular data into four types: (1) **Encoding-based methods** such as VIME (Yoon et al., 2020), SCARF (Bahri et al., 2021), IGTD (Zhu et al., 2021), and SuperTML (Sun et al., 2019); (2) **Novel hybrid architectures** such as DeepFM (Guo et al., 2017), xDeepFM (Lian et al., 2018), and many others Cheng et al. (2016); Frosst & Hinton (2017); Ke et al. (2018; 2019); Popov et al. (2019); Luo et al. (2020); Liu et al. (2020); Ivanov & Prokhorenkova (2021); Luo et al. (2021); (3) **Transformer-based architectures** including SAINT (Somepalli et al., 2021), TabNet (Arik & Pfister, 2021), FT-Transformer Gorishniy et al. (2021), TabTransformer (Huang et al., 2020), and ARM-Net Cai et al. (2021)); (4) **Regularized DNNs** (Shavitt & Segal, 2018; Kadra et al., 2021). For instance, TANGOS introduced special tabular-specific regularization to try to improve deep models' performance (Jeffares et al., 2023). We also note that a more recent work, Tree-hybrid simple MLP (Yan et al., 2024a), proposes to combine MLPs and GBDTs together for the tabular prediction problem.

**Self-supervised learning.** Self-supervised learning (SSL) or unsupervised pretraining has improved performance in settings where there is a significant source of unlabeled data like text (Devlin et al., 2018) and images (Chen et al., 2020). In SSL, deep models are first pretrained on perturbed unlabeled data using self-supervised tasks to learn useful representations for the data. Then these models are finetuned for a downstream task with labeled data. Finetuning often takes one of two forms: (1) linear probing (a popular finetuning strategy in text and images (Chen & He, 2021; Chen et al., 2020; Peters et al., 2019)),where we freeze the pretrained model and only update the linear head during supervised finetuning, and (2) full finetuning, where we update the whole model during supervised finetuning (Devlin et al., 2018). Sometimes a mix of the two is used (Kumar et al., 2022). The core idea behind prominent SSL approaches like SimSiam and SimCLR is to carefully perturb input training samples, and train a representation that is consistent for perturbations of the same sample. This provides robustness to natural perturbations and noise in data (Hendrycks et al., 2019).

Inspired by the success of pretraining and self-supervised learning in images and text, several works show how to apply SSL to unlabeled tabular data. One strategy is to corrupt tabular data and train a deep model to reconstruct it (Yoon et al., 2020; Majmundar et al., 2022; Ucar et al., 2021; Nam et al., 2023b; Lin et al., 2023; Syed & Mirza, 2023; Hajiramezanali et al., 2022). Another approach is to use contrastive losses, which have been highly successful in the image domain (Chen et al., 2020). These methods are also applicable to our case because tabular data, like image data, is often composed of fixed-dimensional real vectors (Verma et al., 2021; Bahri et al., 2021; Lee & Shin, 2022; Hager et al., 2023; Liu et al., 2023; Darabi et al., 2021). Concurrent work has also investigated training tabular-specific LLMs for tabular tasks (Yan et al., 2024b; Schuh et al., 2024). Concurrent to us, Holzmüller et al. (2024) also proposes pretraining (with a modified architecture) to improve tabular deep learning performance. Rubachev et al. (2022) evaluate a variety of different pretraining methods for tabular learning across many different datasets, finding that there is not a clear state of the art.

**Robustness in LTR.** There has been prior work on studying worst-case behavior (robustness) of rankers (Voorhees, 2005; Zhang et al., 2013; Goren et al., 2018; Wu et al., 2022b;a; Penha et al., 2022). Some previous metrics measure a model's robustness against adversarial attack (Goren et al., 2018; Wu et al., 2022a; 2023). Others measure the model's per-query performance variance on a dataset (Voorhees, 2005; Zhang et al., 2013; Wu et al., 2022b). Our outlier metric, Outlier-NDCG, is a departure from previous work because it is not directly a measure of robustness and it is possible for a model to perform *better* on outlier data.

**Transfer learning.** We summarize some related work on transfer learning in tabular domain. One direction revolves around pretraining models on common columns across many datasets (Zhu et al., 2023; Wang &

Sun, 2022; Sui et al., 2023; Ye et al., 2023). Another direction leverages LLMs (large language models) to do few-shot tabular learning (Hegselmann et al., 2023; Liu et al., 2022; Nam et al., 2023a). Pretrained model features are often useful across datasets. For example, text features are transferable between text datasets (Devlin et al., 2018), which has contributed to the recent success of foundation models (Devlin et al., 2018; Touvron et al., 2023; Kojima et al., 2022; Brown et al., 2020). Unfortunately, it is still unclear how to achieve transferability in tabular LTR, where previous work has leveraged common columns (or at least column documentation) to achieve transfer learning (Levin et al., 2022; Hegselmann et al., 2023). However most LTR datasets do not have common columns or publicly-documented column meanings (Qin & Liu, 2013; Chapelle & Chang, 2011; Lucchese et al., 2016). Developing a method to transfer features in tabular LTR is an important open problem.

**Pretrained models in learning to rank.** Concurrent to the submission/review of this paper, recent works have also validated the benefits of pretraining in LTR. S2phere (Li et al., 2023c) proposes an elaborate recipe combining pseudo-labeling, contrastive learning, and transformer-based masked autoencoding to pretrain an LTR model, and shows that it performs well in offline and online metrics. MPGraf (Li et al., 2023b) proposes a pretraining method for LTR that combines both the query-document regression formulation of LTR and the graph-based formulation of LTR into a single model. GS2P (Li et al., 2024) proposes to use an ensemble of differently-trained models to generate pseudo-labels for the final model to learn. Like us, GRACE (Xu et al., 2024) uses contrastive learning for pre-ranking (retrieval); unlike our method, they use this process to align item embeddings with pretrained embeddings during supervised training phase. COLTR (Li et al., 2023d) proposes to increase the dimensionality of the features with Fourier features and to alternative pseudo-labeling between a listwise and pointwise ranker to produce better ranking results. Li et al. (2023a) finetune pretrained language models for the document ranking task.

### A.2 Experimental Details

#### A.2.1 Dataset statistics

In table Table 5, we provide the details of our three public datasets and the proprietary real-world dataset. Note that for the public dataset experiments we sub-sample the pre-training dataset in different ways (as described in Sections 3, 4.1.1 and 4.1.2) to create the training dataset and simulate scarcity of labeled samples. However, we use the same validation dataset without and with the labels for the self-supervised pretraining and supervised learning stages, respectively. Note that in all these datasets, the higher the label value the higher the relevance of the item.

Table 5: Dataset statistics on the MSLRWEB30K, Yahoo Set1, Istella_S, and proprietary datasets.

| | Number of query groups | | | | | Number of | List of |
|---|---|---|---|---|---|---|---|
| | *Pretraining* | | *Supervised learning* | | | | |
| **Dataset** | *Pretrain* | *SSL Val* | *Train* | *Val* | *Test* | **features** | **labels** |
| MSLRWEB30K | 18151 | 6072 | Subset Pretrain | SSL Val | 6072 | 136 | {0,1,2,3,4} |
| Yahoo Set1 | 14477 | 2147 | Subset Pretrain | SSL Val | 5089 | 700 | {0,1,2,3,4} |
| Istella_S | 19200 | 7202 | Subset Pretrain | SSL Val | 6523 | 220 | {0,1,2,3,4} |
| Proprietary dataset | 35.6M | 1.62M | 3.68M | 171k | 1.08M | 63 | {0,1} |

#### A.2.2 Outliers query group details

Here we detail how we select the outlier query groups for the Outlier-NDCG metrics (Section 2.1). For our proprietary online shopping dataset, there are known outlier features that cause low quality predictions. These outliers arise from noise introduced in various stages in the data pipeline. We have mitigation strategies in place against these feature outliers, but they are often imperfect. Because we already know some of the outliers features, we deem any query group with items with outlier features as an outlier. These outliers comprise roughly 10% of the test set, which is still a large number of query groups (much larger than the number of query groups in the outlier datasets of the public datasets).

For our public datasets, we systematically select outliers as follows: we generate a histogram with 100 bins for each feature across the *validation dataset*. For example, Istella (Lamkhede & Kofler, 2021) has 220 features, so we have 220 different histograms. For each histogram, we scan from left to right on the bins until we have encountered at least $G$ empty bins in a row, and if there is less than 1% of the validation set above this bin, then all the feature values above this bin are considered outliers. We also repeat this process right to left. Any test query group containing items with outlier feature values is labeled an outlier query group, and placed in the test outlier dataset. In Figure 5, we present an example histogram of a feature in Istella_S dataset.

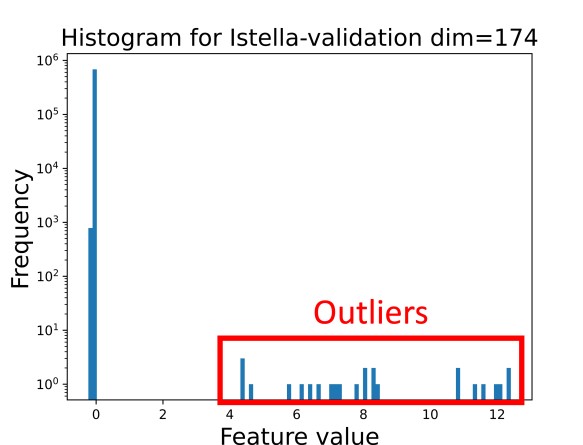

Figure 5: An example of outlier detection in Istella_S (Lucchese et al., 2016) for our Outlier-NDCG metric.

Because different datasets have differently-sized typical gaps, we tune $G$ for each dataset (MSLR, Yahoo, Istella) such that the resulting percentage of outlier queries is as close to 1% of the test set as we can get. MSLRWEB30K has $G = 5$, with 0.65% (40/6072) outlier queries, Yahoo has $G = 20$, with 1.4% (30/2147) outlier queries, and Istella has $G = 32$, with 0.46% (34/7202) outlier queries. 1% is a hyperparameter that can be tuned according to the user's goals.

### A.2.3 Model Settings and Training Details

**Pretrained DL model.**

*Pretraining*: (1) the encoder used for pretraining is the tabular ResNet (Gorishniy et al., 2021) with three ResNet blocks, with the final linear layer removed, (2) pretraining is done on the entire dataset with learning rate 0.0005 using Adam (Kingma & Ba, 2014) (3) with SimSiam/SimCLR-Rank/SimCLR-Sample, we tuned among four different augmentations for each pretraining method: randomly zeroing out features ("Zeroing") with probabilities 0.1 or 0.7, and Gaussian noise with scale 1.0 or 2.0, (4) for SCARF we tune the corruption probability in $\{0.3, 0.6, 0.9\}$, (5) for VIME-self we tune the corruption probability in $\{0.3, 0.5, 0.7\}$, (6) for DACL+ we tune the mixup amount in $\{0.3, 0.6, 0.9\}$, (7) for SubTab we divide input features into 4 subsets with 75% overlap, and tune the masking probability in $\{0.3, 0.6, 0.9\}$, (8) we pretrain for 300 epochs for all methods, and (9) we use a batch size of roughly 200000 items for SimCLR-Rank, SimSiam, VIME-self, and SubTab as they are less GPU memory intensive while using a batch size of roughly 2000 items for SimCLR/DACL+/SCARF which are more GPU memory intensive.

We also evaluate SAINT (Somepalli et al., 2021), a transformer-based pretrainining method. For the architecture, we follow (Somepalli et al., 2021), using $L = 4$ layers, dropout of 0.1, attention heads $h = 4$, with a self-attention query/key/value dimension of 16 and a intersample attention query/key/value dimension of 64. For augmentation, we follow (Somepalli et al., 2021) and use a CutMix mask parameter of 0.3 and MixUp parameter of 0.2. Again following (Somepalli et al., 2021) we weight the denoising loss 10 times more than the contrastive loss. For all other minor architectural details, we follow the code of (Somepalli et al., 2021). We pretrain for 5 epochs with a batch size of roughly 100 items due to GPU memory contraints and

also the time cost (one epoch of saint took 300x longer than an epoch of SimCLR-Rank in MSLRWEB30K, 6000x longer in Yahoo Set1, and 1000x longer in Istella). We use a learning rate of 5e-5.

*Finetuning*: (1) finetuning is done on the labeled train set by adding a three-layer MLP to the top of the pretrained model and training only this head for 100 epochs and then fully finetuning for 100 epochs using Adam with a learning rate of 5e-5, (2) we use an average batch size of roughly 1000 items (may vary based on query group size), (3) we use the LambdaRank loss (Burges, 2010), (4) we use the validation set to perform early stopping (i.e. using the checkpoint that performed best on the validation set to evaluate on the test set).

For SAINT we follow the finetuning process of (Somepalli et al., 2021) in using the output of the CLS embedding as the representation for an item, and tune the learning rate in {5e-5, 1e-5, 5e-6} as we found the performance to be sensitive to this setting.

**Implicit binary user feedback setting.** We reuse the methodology of Section 4.1.1, except we use a scoring head of one linear layer (as opposed to a three layer MLP) for pretrained models. We found that this improved stability and performance in the binary label setting.

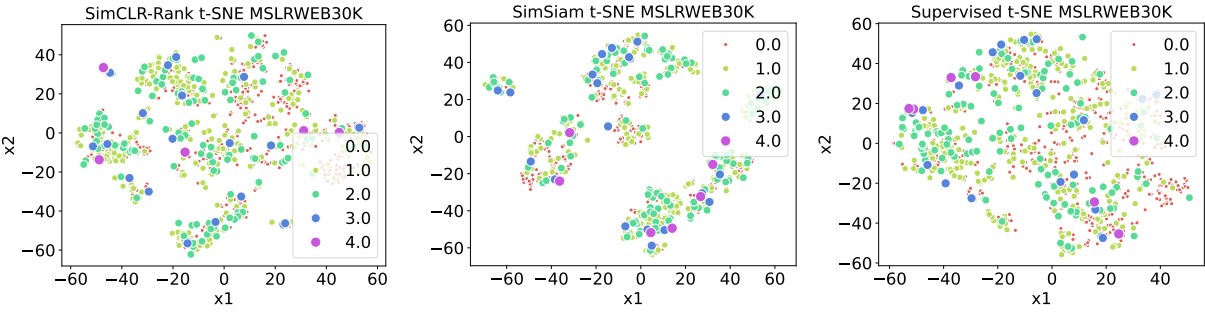

Figure 6: We plot the t-SNE plots of embeddings produced by three different encoders for the MSLRWEB30K dataset: (1) pretrained by SimCLR-Rank, (2) pretrained by SimSiam, (3) trained via supervised training on the entire training set on roughly 1000 samples from MSLR. Marker size and color indicates relevance. We find (1) SimCLR-Rank/SimSiam cluster different relevances effectively but do not order them as well as the supervised encoder. (2) SimCLR-Rank produces more spread-out embeddings with less defined clusters than SimSiam.

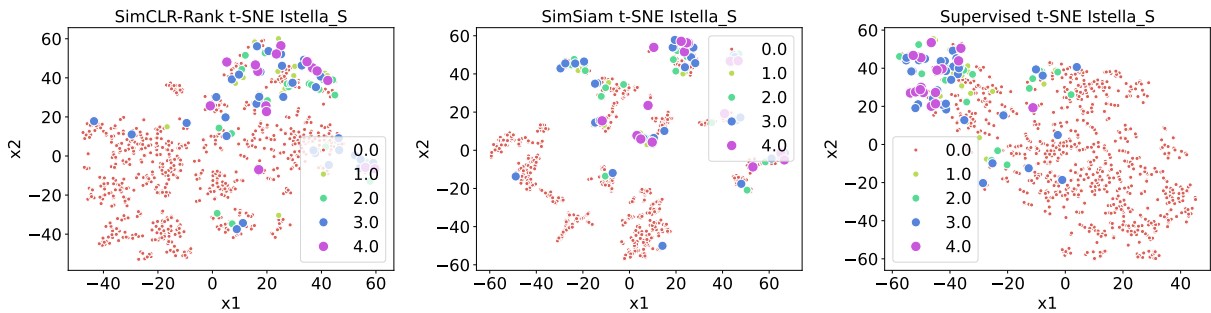

Figure 7: We plot the t-SNE plots of embeddings produced by three encoders: (1) pretrained by SimCLR-Rank, (2) pretrained by SimSiam, (3) trained via supervised training on the entire training set on roughly 1000 samples from Istella_S. Marker size and color indicates relevance. We find (1) SimCLR-Rank/SimSiam cluster different relevances effectively but do not order them as well as the supervised encoder. (2) SimCLR-Rank produces more spread-out embeddings with less defined clusters than SimSiam.

**No Pretrained DL model.** There are a few rankers we evaluated for the no-pretrain DL baseline: (1) a 3-layer tabular ResNet from (Gorishniy et al., 2021) with a three-layer MLP on top of it, and was trained

for 300 epochs on the labeled training set with learning rate tuned in {0.01, 0.001, 0.0001} using Adam, with early stopping using the validation set, (2) a DeepFM (Guo et al., 2017) model following the model architecture in the paper with learning rate tuned in {0.05, 0.005, 0.0005}, and (3) a DCNv2 (Wang et al., 2021) model with learning rate tuned in {0.05, 0.005, 0.0005} and cross layers tuned in {2,3,4}. DeepFM and DCNv2 were designed to use categorical features, and in MSLRWEB30K/Yahoo Set1/Istella_S do not document their categorical features (if any). We performed an analysis of the datasets and found 5, 48, 6 categorical features in MSLRWEB30K/Yahoo Set1/Istella_S respectfully each with cardinality at most 3. We embedded these categorical features into embeddings of dimension 8, following the rule-of-thumb in (Wang et al., 2021).

**GBDT model.** The GBDT ranker is the one from `lightgbm` (Ke et al., 2017) and we grid search the number of leaves in {31, 96, 200} minimum data in leaf in {20, 60, 200}, and column sample in {0.5, 0.9, 1.0} individually for each data point (for a total of 9 difference choices), while letting the rest of the parameters be the default in `lightgbm` (our tuning strategy is similar to Qin et al. (2021)). We use the LambdaRank loss to train the models.

**GBDT + pseudolabeling.** We add pseudo-labeled GBDTs as a semi-supervised GBDT baseline. For GBDTs with pseudo-labeling (GBDT-pseudo), we propose the following algorithm as there are no references to our knowledge. First, we train the GBDT on the labeled train set, and then use this GBDT to pseudo-labels for the unlabeled train set. A GBDT ranker outputs real-valued scores. To convert real-valued scores into relevance scores, we take the resulting scores and evenly bin them into 31 buckets. The scores converted to 0 are the lowest bucket, the scores converted to 1 are the second lowest bucket, and so on. We choose 31 because that is the largest number of labels allowed for GBDT ranking in lightgbm (the reason is because the LambdaRank loss requires the calculation of $2^{\text{label}}$ which can get very large).

**GBDT + PCA.** For GBDT+PCA, we take the entire training set's features and learn a PCA on it with {5, 10, 15} components, similar to (Duh & Kirchhoff, 2008), to produce {5, 10, 15} new features (i.e. we tune among 5, 10, 15). We then train on the labeled train set, with both the original features and the new PCA features together.

**GBDT + PCA + pseudolabeling.** We also introduce a baseline where we perform PCA on the full training dataset, and then perform pseudolabeling as above. We tune the PCA similarly to the GBDT + PCA baseline.

**SimCLR-Rank + GBDT/SimSiam + GBDT.** We also explore using pretrained neural embeddings as features in the GBDT. To do this, we used the pretrained SimCLR-Rank (or SimSiam) model to produce embeddings over the labeled data, concatenated the original features with the embeddings, and then trained a GBDT on this new dataset. We use the same tuning grid as for the original GBDT model.

**Comparing finetuning strategies.** When linear probing (LP), we freeze the pretrained model and update only a linear head on top of it for 200 epochs. In multilayer probing (MP), we freeze the pretrained model and update a 3-layer MLP head on top of it for 200 epochs. In full finetuning (FF), we use the finetuning strategy from Section 4.1.1. We use the validation set for early stopping.

All reported values for Outlier-NDCG are those achieved by the corresponding ranker with best overall NDCG. We report all results as averages over 3 trials, and use single V100 GPUs from a shared cluster to run the experiments.

### A.2.4 Details of Binary Label generation used in Section 4.1.2

Following the methodology from Yang et al. (2022), we generate stochastic binary labels $y$ from the relevance labels $r$ as $y = \mathbb{1}\{t \cdot r + G_1 > t \cdot \tau_{\text{target}} + G_0\}$ where $t$ is a temperature parameter, $G_1, G_0$ are standard Gumbels, and $\tau_{\text{target}}$ is a parameter controlling how sparse the binary labels are. Higher $\tau_{\text{target}}$ leads to higher label sparsity. Yang et al. (2022) show that $y$ is 1 with probability $\sigma(t \cdot (r - \tau_{\text{target}}))$ where $\sigma$ is the sigmoid function. We set $t = 4$ as in Yang et al. (2022). For a given $\tau_{\text{target}}$, we produce produce a new dataset for each of MSLRWEB30K, Yahoo Set1, and Istella_S where we convert the relevance labels in the training/validation sets to binary labels and keep the true relevance labels in the test set.

### A.2.5 Large industry-scale dataset experimental details

For our DL ranker, first we pass the features (numerical and categorical) through some featurizers. Then we use a simple 4 hidden layer MLP, with and ReLU activation, to predict a single scalar score for each (query, item) pair. We also use dropout and batch normalization in input and hidden layers.

For the SimCLR pretraining stage, we remove the last linear layer of the above ranker to create the main encoder to produce the pretrained embeddings. Projector encoder here is a 2 hidden layer MLP, with ReLU activation and no bias, which outputs an input dimensional embedding. We also do not use any normalization or dropout in this projector.

For SimSiam we use the same embedder and a similar projector. The only difference in the projector being batch normalization in the hidden layers and batch normalization without affine learnable parameters in the output layer. Additionaly, SimSiam also employs a predictor MLP, with a single hidden layer and ReLU activation, and with the same input and output sizes. It uses batch normalization and no bias in the hidden layer and no input and hidden dropouts.

For pretraining, we used an augmentation which randomly zeroed out some features. We also tried randomly swapping some features between items in the same batch or query group, but they didn't perform as well as zeros augmentation. During the fine-tuning stage we discard the projectors and predictor and add back the linear layer to complete the non-pretrained ranker structure. DL rankers are trained/fine-tuned using PiRank loss Swezey et al. (2021), batchsize of 1000, AdamW optimizer with learning rate of 0.00001, weight decay of 0.01 and cosine scheduling with warmup. All training stages and experiments were conducted on machines with 4 GPUs, where we ran 100k iterations with ddp and picked the best checkpoint based on validation metric.

GBDT rankers were trained with LambdaRank using the `lightgbm` package. They were tuned with Bayesian hyperparameter optimization strategy by tuning number of leaves, learning rate, and minimum data in leaf in a reasonably large range with a maximum of 150 jobs. We used default values of the library for the other hyperparameters.

## A.3 Additional results

### A.3.1 Runtime comparisons of pre-training strategies

In Table 6, we provide the runtime comparisons between SimSiam, SimCLR, and SimCLR-Rank (new method).

Table 6: Seconds per epoch comparison between pretraining methods. Average over 3 trials. The encoder we use for pretraining is the tabular ResNet (Gorishniy et al., 2021) with the final linear layer taken out.

| Method | Complexity | Seconds per epoch | | |
| --- | --- | --- | --- | --- |
| | | MSLRWEB30K | Yahoo Set1 | Istella |
| SimCLR | $O(B^2 L^2)$ | $102.81 \pm 1.14$ | $14.45 \pm 0.04$ | $69.26 \pm 0.16$ |
| SimCLR-Rank (new) | $O(BL^2)$ | $9.4700 \pm 0.02$ | $1.970 \pm 0.01$ | $5.950 \pm 0.04$ |
| SimSiam | $O(BL)$ | $0.8700 \pm 0.01$ | $0.130 \pm 0.00$ | $0.730 \pm 0.00$ |

### A.3.2 Additional Results on linear probing vs finetuning

Here we provide more results on the question of linear probing vs full finetuning, as discussed in Section 4.3. In Tables 7 and 8 we compare different probing strategies for all datasets when using SimSiam and SimCLR-Rank, respectively, as the pretraining method.

Table 7: SimSiam under different finetuning strategies on NDCG/Outlier-NDCG, averaged over 3 trials (LP = Linear Probing, MP = Multilayer Probing, FF = Full Finetuning). We find that linear probing *and* MLP probing perform extremely poorly (except in Yahoo Set1, where MLP probing performs well).

| Method | MSLRWEB30K ($\uparrow$) | Yahoo Set1 ($\uparrow$) | Istella ($\uparrow$) |
|---|---|---|---|
| | NDCG ($\uparrow$) | | |
| LP | $0.2679 \pm 0.0007$ | $0.6089 \pm 0.0032$ | $0.3805 \pm 0.0042$ |
| MP | $0.2764 \pm 0.0001$ | $\mathbf{0.6137 \pm 0.0022}$ | $0.4484 \pm 0.0020$ |
| FF | $\mathbf{0.3935 \pm 0.0034}$ | $0.6107 \pm 0.0035$ | $\mathbf{0.5618 \pm 0.0049}$ |
| | Outlier-NDCG ($\uparrow$) | | |
| LP | $0.1803 \pm 0.0033$ | $0.5088 \pm 0.002$ | $0.4407 \pm 0.0388$ |
| MP | $0.1749 \pm 0.0023$ | $0.5157 \pm 0.0080$ | $0.5324 \pm 0.0002$ |
| FF | $\mathbf{0.3149 \pm 0.0119}$ | $\mathbf{0.52 \pm 0.0133}$ | $\mathbf{0.6348 \pm 0.0164}$ |

Table 8: SimCLR-Rank under different finetuning strategies on NDCG/Outlier-NDCG, averaged over 3 trials (LP = Linear Probing, MP = Multilayer Probing, FF = Full Finetuning). Multilayer probing performs moderately worse than full finetuning, and linear probing performs much worse than all other strategies.

| Method | MSLRWEB30K ($\uparrow$) | Yahoo Set1 ($\uparrow$) | Istella ($\uparrow$) |
|---|---|---|---|
| | NDCG ($\uparrow$) | | |
| LP | $0.3219 \pm 0.0224$ | $0.5202 \pm 0.0093$ | $0.4029 \pm 0.0073$ |
| MP | $0.3890 \pm 0.0011$ | $0.5942 \pm 0.0016$ | $0.5813 \pm 0.0006$ |
| FF | $\mathbf{0.3959 \pm 0.0022}$ | $\mathbf{0.6022 \pm 0.0013}$ | $\mathbf{0.5839 \pm 0.0013}$ |
| | Outlier-NDCG ($\uparrow$) | | |
| LP | $0.2304 \pm 0.0332$ | $0.3811 \pm 0.0048$ | $0.4717 \pm 0.0166$ |
| MP | $\mathbf{0.2969 \pm 0.0009}$ | $0.4888 \pm 0.0046$ | $\mathbf{0.6369 \pm 0.0045}$ |
| FF | $0.2892 \pm 0.0025$ | $\mathbf{0.5143 \pm 0.0055}$ | $0.6352 \pm 0.0140$ |

### A.3.3   Additional Results on SimCLR-Rank vs SimSiam

In this section we give additional t-SNE plots of the embeddings learned by SimCLR-Rank and SimSiam encoders, and the pre-final layer of the fully supervised ranker (trained with all labels) on MSLRWEB30K dataset (Figure 6), and Istella_S dataset (Figure 7). For the Yahoo Set1 dataset, SimSiam performs the best, but for the above two datasets SimCLR-Rank is the best.

### A.3.4   Additional Results: Simulated crowdsourced label setting

Here we provide the results on label scarcity for Outlier-NDCG in Figure 8. Result interpretation is given in Section 4.1.1.

### A.3.5   Additional Results for Simulated Implicit Binary user feedback setting

In this part of the appendix we provide additional results on binary label generation as detailed in Section 4.1.2. In Tables 9 and 10 we give the dataset statistics when using $\tau_{\text{target}} = 4.25$ and $\tau_{\text{target}} = 5.1$, respectively. For these two settings, the GBDT baseline does not contain semi-supervised runs as we found that semi-supervision tends to generally degrade GBDT performance; see Appendix A.4. The "No Pretrain" baseline uses the tabular ResNet architecture.

In Tables 12 and 13 we give the comparison between pretrained and non-pretrained rankers following the methodology in Section 4.1.2 and find that GBDTs perform the best when binary labels are less sparse ($\tau_{\text{target}} = 4.25$) and pretrained DL rankers perform the best when binary labels are sparser ($\tau_{\text{target}} = 5.1$).

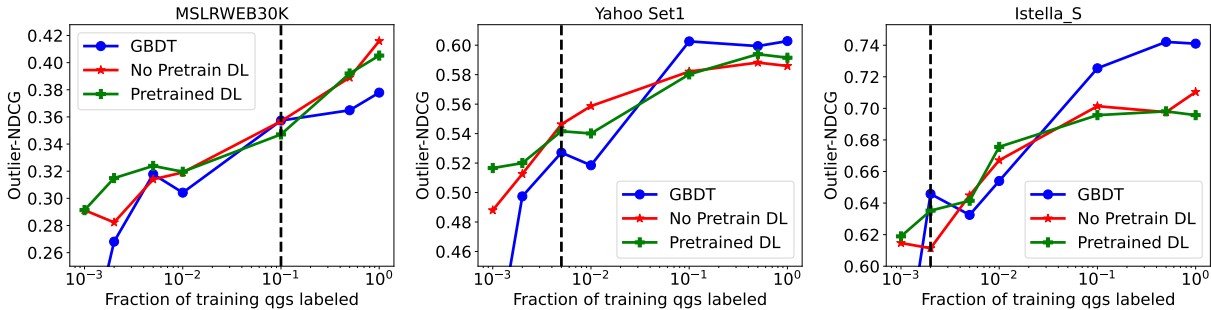

Figure 8: **Simulated crowdsourcing of labels:** We compare Outlier-NDCG ($\uparrow$) pretrained rankers, non-pretrained DL rankers, and GBDT rankers as we change the percentage of training query groups that are labeled. To the left of the black dotted line, pretrained rankers perform the best. Points are averages over three trials. To the left of the black dotted vertical line, pretrained rankers are (1) significantly better on outliers than GBDTs at the $p = 0.05$ level using a one-sided $t$-test, and (2) on average better on outliers than all other non-pretrained methods.

Table 9: Train and validation dataset statistics on MSLRWEB30K, Yahoo Set1, and Istella_S after binary label generation with $\tau_{\text{target}} = 4.25$. The test set is left untouched. Labeled QGs are those query groups that have at least one item with $y = 1$.

| Dataset | # query groups | | # labeled QGs | | # items per QG | | # positives per QG | |
|---|---|---|---|---|---|---|---|---|
| | train | val | train | val | train | val | train | val |
| MSLRWEB30K | 18151 | 6072 | 15.9% | 16.2% | 124.35 | 122.38 | 1.4% | 1.4% |
| Yahoo Set1 | 14477 | 2147 | 11.1% | 10.8% | 30.02 | 30.32 | 6.2% | 6.0% |
| Istella_S | 19200 | 7202 | 49.0% | 49.5% | 106.36 | 94.98 | 1.4% | 1.6% |

This strongly suggests that pretrained DL models outperform GBDTs when the labels arise from binary user feedback that is moderately uncommon or rare (as in online shopping, recommendations, or search).

### A.3.6   Ablation: Data Augmentation Techniques

In Tables 14 and 15, we show the performance of each augmentation choice on SimCLR-Rank and SimSiam, respectively. We use the methodology described in Section 4.1.1, and have 0.2% of training QGs labeled.

### A.3.7   Combining SimCLR-Rank and SimSiam

Here we describe how we constructively combined the SimCLR-Rank and SimSiam to get the competitive results in Section 4.3. Following the setting of Appendix A.2.3, we train (1) a SimCLR-Rank model with gaussian augmentation and noise of scale 1.0, and (2) a SimSiam model with zeroing augmentation and corruption probability 0.1.

After pretraining the SimCLR-Rank and SimSiam models, we finetune each of them on the labeled data and keep the checkpoints with the best validation NDCG. Then we combine these two models by concatenating the final embedding layers together—if each model's penultimate embedding dimension is $h$ ($h = 136$ for us), then the concatenated embedding is of dimension $2 \times h$. Then we pass this concatenated model through a batchnorm (to normalize the scales of each embedding to prevent scale issues), pass it through dropout layer of probability 0.7 (if the dataset is MSLRWEB30K or Istella_S) or 0.0 (if the dataset is Yahoo Set1), and map this vector (of dimension $2 \times h$) to a score of dimension 1 using a linear layer. We finetune only the final linear layer.

Table 10: Train and validation dataset statistics on MSLRWEB30K, Yahoo Set1, and Istella_S after binary label generation with $\tau_{\text{target}} = 5.1$. The test set is left untouched. Labeled QGs are those query groups that have at least one item with $y = 1$.

| Dataset | # query groups | | # labeled QGs | | # items per QG | | # positives per QG | |
|---|---|---|---|---|---|---|---|---|
| | train | val | train | val | train | val | train | val |
| MSLRWEB30K | 18151 | 6072 | 1.1% | 1.3% | 124.35 | 122.38 | 0.8% | 0.7% |
| Yahoo Set1 | 14477 | 2147 | 0.6% | 0.6% | 30.02 | 30.32 | 5.4% | 5.1% |
| Istella_S | 19200 | 7202 | 3.0% | 2.7% | 106.36 | 94.98 | 1.1% | 1.3% |

Table 11: Train and validation dataset statistics on MSLRWEB30K, Yahoo Set1, and Istella_S after binary label generation with $\tau_{\text{target}} = 4.5$. The test set is left untouched. Labeled QGs are those query groups that have at least one item with $y = 1$.

| Dataset | # query groups | | # labeled QGs | | # items per QG | | # positives per QG | |
|---|---|---|---|---|---|---|---|---|
| | train | val | train | val | train | val | train | val |
| MSLRWEB30K | 18151 | 6072 | 8.9% | 8.9% | 124.35 | 122.38 | 1.1% | 1.1% |
| Yahoo Set1 | 14477 | 2147 | 5.3% | 5.4% | 30.02 | 30.32 | 5.9% | 5.4% |
| Istella_S | 19200 | 7202 | 25.2% | 25.2% | 106.36 | 94.98 | 1.3% | 1.3% |

## A.4 Semi-supervised GBDT

Here we provide a more detailed comparison between Supervised GBDTs, GBDT + pseudolabeling, and GBDT + PCA + pseudolabeling in Figure 9 for the simulated crowdsourcing of labels setting and in Table 17 for the simulated implicit binary user feedback setting. We follow the experimental methodology in Appendix A.2.3. We find that for the most part, semi-supervised learning does not benefit GBDT rankers.

## A.5 Quantitative measures of embedding quality

We provide a quantitative measure of embedding quality in this section. We attempt to verify that the focus on in-query group negatives helps improve embedding quality. To do this, we compare the embeddings produced by a SimCLR-Rank model and a SimCLR-Sample model (SimCLR-Sample is the same as SimCLR-Rank except the negatives are randomly taken from the batch rather than being in-query group), both under zeroes augmentation with $p = 0.1$, and measure how spread out the embeddings are within a query group for each method. In Table 16, we find that SimCLR-Rank embeddings are consistently more spread out within a query group than SimCLR-Sample, showing that in-query group focus leads to a measurable impact on the distinguish-ability of embeddings within a query group. This may help explain why SimCLR-Rank performs better than SimCLR-Sample.

Table 12: We compare pretrained models to non-pretrained models in the binary label setting with $\tau_{\text{target}} = 4.25$ (Section 4.1.2) on NDCG averaged over three trials. We follow the methodology in Section 4.1.2.

| Method | MSLRWEB30K (↑) | Yahoo Set1 (↑) | Istella (↑) |
|---|---|---|---|
| | NDCG (↑) | | |
| GBDT | $0.3616 \pm 0.0000$ | $0.6233 \pm 0.0000$ | $0.6251 \pm 0.0000$ |
| No Pretrain DL | $0.3552 \pm 0.0015$ | $0.6297 \pm 0.0005$ | $0.6147 \pm 0.0006$ |
| Pretrained DL | $0.3602 \pm 0.0007$ | $0.6272 \pm 0.0024$ | $0.6243 \pm 0.0007$ |
| | Outlier-NDCG (↑) | | |
| GBDT | $0.2792 \pm 0.0000$ | $0.5352 \pm 0.0000$ | $0.7393 \pm 0.0000$ |
| No Pretrain DL | $0.3142 \pm 0.0063$ | $0.5389 \pm 0.0036$ | $0.6631 \pm 0.0102$ |
| Pretrained DL | $0.2923 \pm 0.0079$ | $0.5471 \pm 0.0050$ | $0.6945 \pm 0.0041$ |

Table 13: We compare pretrained models to non-pretrained models in the binary label setting with $\tau_{\text{target}} = 5.1$ (Section 4.1.2) on NDCG averaged over three trials. We follow the methodology in Section 4.1.2. ♣ indicates metrics on which pretrained rankers outperform non-pretrained rankers significantly via a $p < 0.05$ t-test.

| Method | MSLRWEB30K | Yahoo Set1 | Istella |
|---|---|---|---|
| | NDCG (↑) | | |
| GBDT | $0.2844 \pm 0.0000$ | $0.5782 \pm 0.0000$ | $0.5638 \pm 0.0000$ |
| No Pretrain DL | $0.3432 \pm 0.0065$ | $0.5703 \pm 0.0246$ | $0.5722 \pm 0.0080$ |
| Pretrained DL | $\mathbf{0.3564 \pm 0.0054}$♣ | $\mathbf{0.6031 \pm 0.0064}$ ♣ | $\mathbf{0.599 \pm 0.0014}$ ♣ |
| | Outlier-NDCG (↑) | | |
| GBDT | $0.2416 \pm 0.0000$ | $0.5011 \pm 0.0000$ | $0.6640 \pm 0.0000$ |
| No Pretrain DL | $0.2441 \pm 0.0089$ | $0.4593 \pm 0.0317$ | $0.6334 \pm 0.0054$ |
| Pretrained DL | $\mathbf{0.2553 \pm 0.0161}$ | $\mathbf{0.54 \pm 0.0055}$♣ | $\mathbf{0.6730 \pm 0.0090}$ ♣ |

Table 14: We compare different augmentation strategies for SimCLR-Rank. The methodology used is the one in Section 4.1.1, with 0.2% of training QGs labeled.

| Augmentation | MSLRWEB30K | Yahoo Set1 | Istella |
|---|---|---|---|
| | NDCG (↑) | | |
| Zeroing (p=0.1) | $0.3830 \pm 0.0007$ | $\mathbf{0.6022 \pm 0.0013}$ | $0.5820 \pm 0.0024$ |
| Zeroing (p=0.7) | $0.3737 \pm 0.0025$ | $0.5998 \pm 0.0053$ | $0.5646 \pm 0.0048$ |
| Gaussian noise (scale=1.0) | $\mathbf{0.3959 \pm 0.0022}$ | $0.5998 \pm 0.0026$ | $\mathbf{0.5839 \pm 0.0013}$ |
| Gaussian noise (scale=2.0) | $0.3907 \pm 0.0025$ | $0.5953 \pm 0.0043$ | $0.5809 \pm 0.0010$ |
| | Outlier-NDCG (↑) | | |
| Zeroing (p=0.1) | $0.2782 \pm 0.0036$ | $\mathbf{0.5143 \pm 0.0055}$ | $\mathbf{0.6408 \pm 0.0063}$ |
| Zeroing (p=0.7) | $0.2730 \pm 0.0054$ | $0.5062 \pm 0.0070$ | $0.6141 \pm 0.0064$ |
| Gaussian noise (scale=1.0) | $\mathbf{0.2892 \pm 0.0025}$ | $0.4963 \pm 0.0024$ | $0.6352 \pm 0.0140$ |
| Gaussian noise (scale=2.0) | $0.2886 \pm 0.0096$ | $0.4875 \pm 0.0054$ | $0.6327 \pm 0.0191$ |

Table 15: We compare different augmentation strategies for SimSiam. The methodology used is the one in Section 4.1.1, with 0.2% of training QGs labeled.

| Augmentation | MSLRWEB30K | Yahoo Set1 | Istella |
|---|---|---|---|
| | NDCG (↑) | | |
| Zeroing (p=0.1) | **0.3935 ± 0.0034** | **0.6107 ± 0.0035** | 0.5618 ± 0.0049 |
| Zeroing (p=0.7) | 0.3911 ± 0.0003 | 0.6076 ± 0.0072 | **0.5660 ± 0.0047** |
| Gaussian noise (scale=1.0) | 0.3782 ± 0.0093 | 0.6010 ± 0.0026 | 0.5612 ± 0.0060 |
| Gaussian noise (scale=2.0) | 0.3860 ± 0.0011 | 0.6100 ± 0.0089 | 0.5587 ± 0.0047 |
| | Outlier-NDCG (↑) | | |
| Zeroing (p=0.1) | **0.3149 ± 0.0119** | **0.5200 ± 0.0133** | **0.6348 ± 0.0164** |
| Zeroing (p=0.7) | 0.3002 ± 0.0060 | 0.5081 ± 0.0097 | 0.6201 ± 0.0104 |
| Gaussian noise (scale=1.0) | 0.2585 ± 0.0371 | 0.4893 ± 0.0084 | 0.6145 ± 0.0160 |
| Gaussian noise (scale=2.0) | 0.2929 ± 0.0021 | 0.5163 ± 0.0101 | 0.5905 ± 0.0146 |

Table 16: The ratio of the in-query group mean absolute deviation over the overall mean absolute deviation of embeddings from SimCLR-Rank and SimCLR-Sample. These embeddings were calculated over the validation set of each dataset. We find that SimCLR-Rank has a higher ratio, meaning the embeddings are more spread out within a query group.

| Method | MSLRWEB30K | Yahoo Set1 | Istella |
|---|---|---|---|
| SimCLR-Rank | 0.9953 | 0.9904 | 0.9915 |
| SimCLR-Sample | 0.9945 | 0.9881 | 0.9797 |

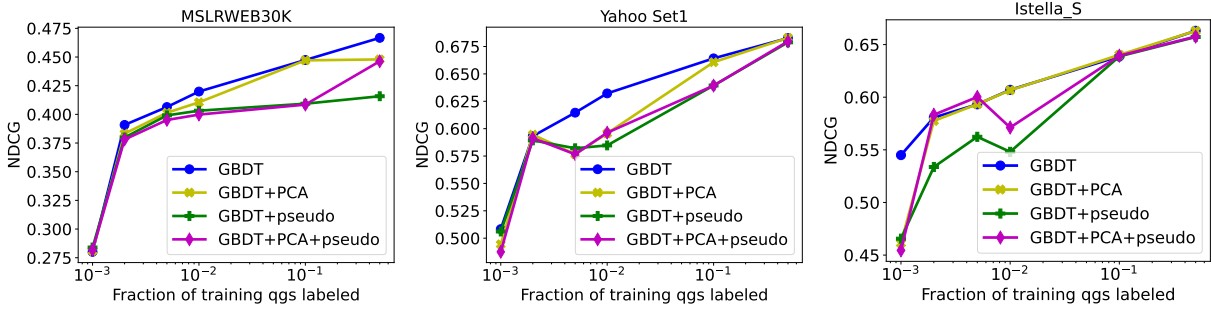

Figure 9: A comparison of Supervised GBDT, GBDT + PCA, GBDT + pseudolabeling, and GBDT + PCA + pseudolabeling. We find that semi-supervised learning may sometimes benefit GBDT rankers, though it is not very consistently beneficial.

Table 17: We compare GBDT models in the binary label setting with $\tau_{\text{target}} = 4.5$ (Section 4.1.2) on NDCG averaged over three trials. We follow the methodology in Section 4.1.2.

| Method | MSLRWEB30K | Yahoo Set1 | Istella |
|---|---|---|---|
| | NDCG (↑) | | |
| Supervised GBDT | 0.3335 ± 0.0000 | 0.6168 ± 0.0000 | 0.6024 ± 0.0000 |
| GBDT + PCA | 0.3105 ± 0.0236 | 0.6141 ± 0.0022 | 0.5508 ± 0.0410 |
| GBDT + pseudolabeling | 0.3467 ± 0.0011 | 0.5868 ± 0.0012 | 0.5178 ± 0.0068 |
| GBDT + PCA + pseudolabeling | 0.3241 ± 0.0270 | 0.5944 ± 0.0012 | 0.5315 ± 0.0120 |

