# OpenReview forum: "Pretrained deep models outperform GBDTs in Learning-To-Rank under label scarcity"
_TMLR — Accepted by TMLR_

### Review · Reviewer_7Esm · 2024-07-12

**Summary Of Contributions:**

This paper is at its core a straightforward empirical study comparing two families of methods in the setting of a particular learning task, namely the task of ranking, where the quality of ranking is evaluated using "normalized discounted cumulative gain" (NDCG), which assumes relevance "labels" associated with each feature vector. More precisely, the authors are interested in learning scenarios in which labels (relevance scores upon which the "ideal" ordering is assumed to be determined) are *scarce*, i.e., a large fraction of the feature vectors observed are not paired with labels. Furthermore, the data is assumed to be "tabular data," i.e., the feature vectors are not assumed to have any special structure as is found in for example image or text data.

The two families of methods being compared are, roughly speaking, gradient-boosted decision trees (GBDT) and deep neural network models. Since most of the data is assumed to be unlabeled, each "method" here is really a pair, namely a model and a pre-training strategy. The context for this paper is that for tabular data, the general consensus based on the existing literature is that GBDTs outperform deep neural networks. This work positions itself as having identified a natural and important learning scenario in which neural networks tend to be superior. In particular, they show that on certain idiosyncratic "outlier" data points, pre-training NNs is a particularly effective strategy.

Their main results are essentially empirical analysis which elucidates effective strategies for using NNs to beat GBDTs in the scenario of interest, though the paper includes a ranking-specific pre-training technique, which adapts the core idea of the well-known SimCLR to the tabular data setting here.

**Audience:**

Yes

**Broader Impact Concerns:**

Not applicable.

**Claims And Evidence:**

Yes

**Requested Changes:**

I don't have any critical questions, just a handful of points I tripped up on while reading.

- NDCG is given as an acronym everywhere in the paper except in the expositions of "contributions" early in the paper. Please define it again near the formal definition in equation (1).
- The authors use the symbol "($\\uparrow$)" throughout the paper; does this mean "larger is better"? This notation is not something I am familiar with; perhaps it is common in applied research papers, but I think a proper definition should be given at the time of the first occurrence in the paper.
- After equation (1), it says *"...is a ranking of the $L$ elements induced by the scoring function $f\_{\\theta}$ on $\\{\\boldsymbol{x}\_{i}\\}\_{i=1}^{L\_{k}}$..."*, but this $L\_{k}$ should be just $L$, right?
- I find the formulation of the data in the ranking problem a bit sub-standard. There are several points which I think need revision:
  - The authors talk about "query groups," but the key formulation in section 2 doesn't make this concrete. There is only one query group present in section 2, correct? Is the notion of a "query" even relevant here? I get the intention of course, and I assume this is standard jargon in this domain, but in the present paper, it is just undefined jargon.
  - When we have multiple query groups, how is evaluation done? Are NDCG values just averaged over query groups? I think this should be formulated in section 2, since later on pre-training techniques with a "batch" of $B$ query groups appear.
  - The "labels are scarce" bit is a key characteristic of the problem being studied here, and yet it is not formalized in section 2. We have $L$ labels and $L$ feature vectors; formalizing the unlabeled data points in section 2 seems far more natural to me.
- Regarding the basic "hypothesis" underlying this work, the authors say *"We hypothesize that pretrained deep models can beat GBDTs in settings that more accurately reflect real-world challenges."* Upon what grounds do the authors make this hypothesis? Previous work is said to study "idealized problem settings," but it is not clear to the reader why we would expect DNNs to be any better than GBDTs in such a setting, given the strong empirical track record of GBDTs on tabular data.

**Strengths And Weaknesses:**

__Strengths:__

The goals of this research work are clearly described in the context of the existing literature, and the main results are presented with clear evidence. Whether the empirical results are surprising or not is not something I can comment on with confidence, but there appears to be a clear take-away in terms of an effective neural network-based strategy for learning to rank under scarce labels.


__Weaknesses:__

Overall, I do not have any major complaints. For better or for worse, this is applied machine learning research, running numerical tests to study the validity of a particular hypothesis. I personally did not find the results particularly surprising, due to the plethora of techniques for effectively pre-training neural networks (in contrast with GBDTs), but that does not invalidate the authors' results, which are likely to be of value for practitioners interested in the ranking problem using tabular data under scarce labels.

---

> ### Author Response · Authors · 2024-07-23
> **Thank you for the thorough feedback!**
>
> We thank the reviewer for their work in writing this comprehensive review, which contains many helpful comments and suggestions. We are encouraged that the reviewer found that our work would “be of value for practitioners interested in the ranking problem using tabular data under scarce labels”.
>
> We would like to comment on a few points (denoted P1, P2,...) made in the review:
>
> **P1**: “For better or for worse, this is applied machine learning research, running numerical tests to study the validity of a particular hypothesis.”
>
> The reviewer is correct: our paper is about running experiments to determine whether pretraining can enable deep models to outperform GBDTs in the presence of label scarcity. Testing hypotheses is at the core of how we advance knowledge in empirical sciences.
>
> **P2**: “I personally did not find the results particularly surprising, due to the plethora of techniques for effectively pre-training neural networks (in contrast with GBDTs)”
>
> The reviewer is right that there are many techniques for pre-training neural networks. At the same time, we would also like to highlight that all prior attempts at using neural networks for tabular data (with or without pretraining) have either not compared against GBDTs or failed to outperform them successfully [3,4]. *The main novelty in our paper is that we diverge from this trend*, backed by our thorough experimental evaluation. Note that our GBDT baselines also include non-trivial semi-supervised techniques for utilizing unlabeled data (PCA, pseudolabeling) which prior works did not evaluate against. It is not obvious a priori pretrained deep models would beat these carefully crafted baselines.
>
> An additional novelty in our paper  is that we found that simple pre-training strategies often perform better than more complicated “tabular-specific” pretraining strategies proposed in prior work [5,6]. We believe this is a surprising and useful result.
>
>
> **P3**: “Regarding the basic "hypothesis" underlying this work, the authors say "We hypothesize that pretrained deep models can beat GBDTs in settings that more accurately reflect real-world challenges." Upon what grounds do the authors make this hypothesis? Previous work is said to study "idealized problem settings," but it is not clear to the reader why we would expect DNNs to be any better than GBDTs in such a setting, given the strong empirical track record of GBDTs on tabular data.”
>
> Thank you for your comment! This is very helpful feedback–we should have made the rationale more clear.
>
> The basis for the hypothesis is as follows. In prior work, pretrained deep models have displayed strong robustness to noise [1] and also a remarkable ability to learn from few labeled examples [2]. This motivated us to propose a non-trivial hypothesis that pretrained deep models can also do the same for tabular data and in tabular LTR.
>
> We will include this explanation in the next revision of the paper.
>
> On the other writing comments: we thank the reviewer and agree with these comments. We will address them at the end of the rebuttal process, as recommended by the TMLR author guidelines.
>
>
> ## Citations
> [1] Dan Hendrycks, Mantas Mazeika, Saurav Kadavath, and Dawn Song. Using self-supervised learning can improve model robustness and uncertainty. Advances in neural information processing systems, 32, 2019.
> [2] Brown, Tom, et al. "Language models are few-shot learners." Advances in neural information processing systems 33 (2020): 1877-1901.
> [3] Gorishniy, Yury, et al. "Revisiting deep learning models for tabular data." Advances in Neural Information Processing Systems 34 (2021): 18932-18943.
> [4] Qin, Zhen, et al. "Are neural rankers still outperformed by gradient boosted decision trees?." International Conference on Learning Representations. 2021.
> [5] Yoon, Jinsung et al. “VIME: Extending the Success of Self- and Semi-supervised Learning to Tabular Domain.” Neural Information Processing Systems (2020).
> [6] Ucar, Talip, Ehsan Hajiramezanali, and Lindsay Edwards. "Subtab: Subsetting features of tabular data for self-supervised representation learning." Advances in Neural Information Processing Systems 34 (2021): 18853-18865.

---

> > ### Author Response · Authors · 2024-09-09
> > **Paper revised**
> >
> > We would like to inform the reviewer that we have now updated the presentation of our hypothesis in Section 1 and the problem formulation in Section 2 as suggested by the reviewer.

---

> > > ### Author Response · Authors · 2024-09-20
> > >
> > > Dear Reviewer 7Esm,
> > >
> > > Thank you again for your thorough review and valuable feedback. We just wanted to check if you had any additional questions or concerns that we can address. Thank you for your consideration!
> > >
> > > Best regards,
> > > The Authors

---

### Review · Reviewer_DqZd · 2024-08-05

**Summary Of Contributions:**

This paper explores the use of unsupervised pretraining for deep learning models in tabular learning-to-rank (LTR) tasks, particularly in settings with scarce labeled data.

The authors demonstrate that these models can outperform gradient boosted decision trees (GBDTs), currently regularly considered state-of-the-art for tabular data, in LTR tasks when labels are relatively scarce. They introduce a new LTR-specific pretraining method called SimCLR-Rank and evaluate various pretraining strategies on both public LTR datasets and a large-scale proprietary dataset. Overall, the paper argues that pretrained deep models can achieve better performance (especially against outliers), and challenges the current understanding of optimal modeling choices for tabular learning.

**Audience:**

Yes

**Broader Impact Concerns:**

C1: It would be good to see a deeper analysis of why the proposed DL methods work well in this context.

For instance:

1. Does the contrastive learning objective related well to learning ranking functions? Are the learned representations capturing important notions in these domains? (e.g. relevance, preference)
2. In what way does the data augmentation + contrastive learning help with the limited data? How does the model actually change under the constraint and additional data?
3. Does the focus on whitin-query comparisons lead to more informative representation for ranking? If so, is there a qualitative / quantitative difference that can be observed by probing the model?

C2: section 3 does perhaps too terse of a job at introducing the methods the general ML reader in the context of tabular LTR.

For instance, semi-supervised learning for GBDTs and pretraining methods for tabular data are introduced without a smooth transition or clear connection between these two concepts.

A straightforward way to address this would be to add a discussion on the motivation for pretraining methods for deep models in LTR, after discussing the limitations of semi-supervised approaches for GBDTs, but the section could also be overhauled to provide a top to bottom story that connects the methods.

Simiarly, SimCLR and SimSiam are also introduced without particularly delving into why exploring domain-agnostic methods is important in this setting. i.e. somewhere in this sesction or beforehand it would be valuable to start convincing the reader that the core idea of learning similar representation for augmented data may generalize across domains. It would also be great to see a paragraph or two on what makes the standard vanilla SimCLR loss limited for the new setting.

These are just a few examples. Overall section 3 could use a rewrite so that the context of the model design choices are well communicated wrt. the problem and the relevant limitation of the baselines.

C3: there's a fair amount of work in deep learning for tabular data (and even just transformer-based approaches) -- see https://arxiv.org/abs/2110.01889. It would be good to see a discussion of these, and perhaps an attempt at evaluating some of the more popular methods in the literature in the setting.

**Claims And Evidence:**

Yes

**Requested Changes:**

W1: Some experimental choices and analyses don't seem to be well-justified. For example, the rationale behind choosing specific architectures for pretrained and non-pretrained models is unclear to me. It is not evident why DeepFM and DCNv2 were not evaluated with pretraining, and if that'd make a significant difference.

W2: It is mechanistically difficult to reationalize how SimCLR-Rank, SimSiam, and the combined model work. This is especially striking due to the authors' observation that these models seem to have a high variance in performance depending on which task they get trained on.

W3: There's generally a lack of analysis over the inductive biases of these methods and how they align with the characteristics of tabular LTR data. The methods are often simply introduced and motivated due to their success in other domains, but it's not clear to me what led to them specifically being applied.

W4: it's not totally clear from the manuscript how SimCLR-Rank compare against other somewhat recent neural tabular learning methods like SAINT, TabNet, etc. -- The manuscript seems to be trying to walk the line between making a general statement about DL methods in the setting, and raising the profile of the proposed method; but the former requires a more thorough analysis of existing methods for the statement to generalize.

**Strengths And Weaknesses:**

S1: LTR with limited labels is an [unfortunately] common situation in real-world applications. This work bridges a gap in the literature by demonstrating a setting where deep learning models seem to consistently outperform GBDTs on tabular data. Challenges the status quo are extremely valuable, and I commend the authors for having a go at this one in particular.

S2: SimCLR-Rank has a clever architecture--specifically the grouping of items within queries is an interesting ways of improving constrastive learning. That seems to me a very effective way of improving on the representation learning problem in these settings, and it's the kind of method that relies on the implicit structure of the data (and thus might generalize effectively in real settings).

S3: The authors conduct extensive experiments on multiple public datasets (`MSLRWEB30K`, `Yahoo Set1`, `Istella_S`) as well as a large-scale (though unfortunately proprietary) dataset, providing a comprehensive evaluation of their approach. The experiments cover various scenarios (simulated crowdsourcing, implicit binary user feedback, etc.), and that gives a lot of strength to the narrative of the manuscript. Furthermore the thorough analysis of the proposed methods with plenty of ablation studies over different pretraining / finetuning approaches is very good to see.

---

> ### Author Response · Authors · 2024-09-09
> **Thank you for insightful questions --- part 1**
>
> We thank the reviewer for their thoughtful comments and helpful suggestions.
>
> > W1: The rationale behind choosing specific architectures for pretrained and non-pretrained models is unclear to me. It is not evident why DeepFM and DCNv2 were not evaluated with pretraining, and if that'd make a significant difference.
>
> Our goal was not to conduct a comprehensive exploration of base architectures to make pretraining effective. In fact, our goal was somewhat the opposite: we wanted to show that even without carefully selecting the base LTR architecture, pretraining can significantly impact results. We feel that conducting a comprehensive study of architectures is a separate and important question: indeed, it is not even clear to us that existing architectures designed to perform well on tabular architectures will also be the best architectures with pretraining. We will highlight this point as an important direction for future work.
>
> > W2: It is mechanistically difficult to reationalize how SimCLR-Rank, SimSiam, and the combined model work. This is especially striking due to the authors' observation that these models seem to have a high variance in performance depending on which task they get trained on.
>
> We agree with your point—we have tried multiple visualizations to try to explain these effects, and so far the best results have come from examining t-SNE visualizations of embeddings by the different methods. These visualizations suggest that for a given dataset and problem setting, the best pretraining method in downstream performance appears to also better cluster the embeddings by relevance. This is discussed further in Figure 4.
>
> > W3: There's generally a lack of analysis over the inductive biases of these methods and how they align with the characteristics of tabular LTR data. The methods are often simply introduced and motivated due to their success in other domains, but it's not clear to me what led to them specifically being applied.
>
> Qualitatively, the ranking relevance of tabular records to a query is typically determined by a combination of fields. For example, if a person searches for a product (say a pair of sandals), we can think of the fields in the LTR problem as representing properties of the results, such as material, color, price, etc. Qualitatively, we expect that even if a record is missing some of these fields (i.e., the zero-augmentation strategy we use in our paper), if the other fields are the same, then the product should usually remain relevant to the user’s query. Indeed, many recommendation systems in e-Commerce websites operate by presenting users with products that have similar field values to a product they already examined. The pretraining methods we examine take advantage of of this insight to force the embeddings of similar tabular records (in semantic space) closer together in vector space. We have added such a discussion to Section 3.
>
> > W4: it's not totally clear from the manuscript how SimCLR-Rank compare against other somewhat recent neural tabular learning methods like SAINT, TabNet, etc.
>
> Thank you for your suggestion. We have run additional experiments comparing against SAINT. We did not compare against TabNet because SAINT already compares against TabNet in [1] and finds that SAINT outperforms TabNet. The results are included below. We started with the default recommended hyperparameters for SAINT, and tuned its learning rate.
>
> | **Method**               | **MSLR (↑)**        | **Yahoo Set1 (↑)**      | **Istella (↑)**        |
> |--------------------------|---------------------|-------------------------|------------------------|
> | SAINT (Somepalli et al., 2021)| 0.3355 ± 0.0043 | 0.5890 ± 0.0075          | 0.5560 ± 0.0066     |
> | SimCLR-Rank               | **0.3868 ± 0.0026** | 0.5843 ± 0.0062          | 0.5609 ± 0.0040        |
> | SimSiam (Chen & He, 2021) | 0.3790 ± 0.0028     | **0.6100 ± 0.0072**      | 0.5189 ± 0.0096   |
>
> The results indicate that SAINT does not outperform the simpler pretraining methods we focused on in our extended evaluation. We have added these results to Table 1 in the manuscript.
>
> [1] Somepalli, G., Schwarzschild, A., Goldblum, M., Bruss, C.B. and Goldstein, T., SAINT: Improved Neural Networks for Tabular Data via Row Attention and Contrastive Pre-Training. In NeurIPS 2022 First Table Representation Workshop.

---

> > ### Author Response · Authors · 2024-09-09
> > **Thank you for insightful questions --- part 2**
> >
> > > C1: It would be good to see a deeper analysis of why the proposed DL methods work well in this context.
> >
> > > Does the contrastive learning objective related well to learning ranking functions? Are the learned representations capturing important notions in these domains? (e.g. relevance, preference)
> >
> > To provide some understanding, in Figure 4 we have added t-SNE plots that compare different pretraining methods while keeping the augmentation strategy fixed (as suggested by Reviewer fFBs. For that dataset (Yahoo), we expect SimSiam to perform best. Qualitatively, we see that the SimSiam embeddings are the most clustered for the relevance classes with the highest weight in the NDCG metric (namely, Relevance labels 3 and 4). We hypothesize that this clustering effect is helping the final ranker separate embeddings, and produce better-quality rankings.
> >
> > > In what way does the data augmentation + contrastive learning help with the limited data? How does the model actually change under the constraint and additional data?
> >
> > We didn’t fully understand this question. Could you please elaborate on what you meant?
> >
> > > Does the focus on whitin-query comparisons lead to more informative representation for ranking? If so, is there a qualitative / quantitative difference that can be observed by probing the model?
> >
> > We ran an experiment which compares the average distance of the item embedding from the mean embedding of all items in the same QG. For normalizing the scale we also divided it by the average distance of item embeddings from dataset-level mean embedding. We evaluate this normalized distance metric for SimCLR-Rank (which takes as negatives all other items from the same QG) and the baseline SimCLR-Sample (which samples equal number of negatives from the whole batch, potentially from different QGs).
> >
> > | **Method**       | **MSLRWEB30K** | **Yahoo Set1** | **Istella** |
> > |------------------|----------------|----------------|-------------|
> > | SimCLR-Rank      | 0.9953         | 0.9904         | 0.9915      |
> > | SimCLR-Sample    | 0.9945         | 0.9881         | 0.9797      |
> >
> > Our findings show that by focusing on negatives within the QG (i.e., SimCLR-Rank), we can increase the separation between items within the QG. This suggests that the embeddings of SimCLR-Rank may be easier to separate than those of SimCLR-Sample by the final linear layer, perhaps explaining the superiority of SimCLR-Rank over SimCLR-Sample. We have included these results in Appendix A.5.
> >
> > > C2: section 3 does perhaps too terse of a job at introducing the methods the general ML reader in the context of tabular LTR.
> >
> > > For instance, semi-supervised learning for GBDTs and pretraining methods for tabular data are introduced without a smooth transition or clear connection between these two concepts.
> >
> > > A straightforward way to address this would be to add a discussion on the motivation for pretraining methods for deep models in LTR, after discussing the limitations of semi-supervised approaches for GBDTs, but the section could also be overhauled to provide a top to bottom story that connects the methods.
> >
> > > Simiarly, SimCLR and SimSiam are also introduced without particularly delving into why exploring domain-agnostic methods is important in this setting. i.e. somewhere in this sesction or beforehand it would be valuable to start convincing the reader that the core idea of learning similar representation for augmented data may generalize across domains. It would also be great to see a paragraph or two on what makes the standard vanilla SimCLR loss limited for the new setting.
> >
> > > These are just a few examples. Overall section 3 could use a rewrite so that the context of the model design choices are well communicated wrt. the problem and the relevant limitation of the baselines.
> >
> > Thank you for the suggestions, we have revised Section 3 accordingly.
> >
> > > C3: there's a fair amount of work in deep learning for tabular data (and even just transformer-based approaches) -- see https://arxiv.org/abs/2110.01889. It would be good to see a discussion of these, and perhaps an attempt at evaluating some of the more popular methods in the literature in the setting.
> >
> > Thank you for the suggestion. We have updated the related work to include these works, and we have added experiments comparing against a leading representative Transformer-based method (SAINT). We did not compare against TabNet because SAINT already demonstrated that it outperforms TabNet. The added experiments and discussion of related work do not significantly change our story.

---

> > > ### Author Response · Authors · 2024-09-20
> > >
> > > Dear Reviewer DqZd,
> > >
> > > Thank you again for your thorough review and valuable feedback. We just wanted to check if you had any additional questions or concerns that we can address. Thank you for your consideration!
> > >
> > > Best regards,
> > > The Authors

---

### Review · Reviewer_fFBs · 2024-08-26

**Summary Of Contributions:**

In this work, the authors present empirical results which suggest that deep learning (DL) models can outperform gradient-boosted decision trees (GBDTs) on the task of learning-to-rank (LTR) on tabular data when the labels are sparse. This contrasts with the standard view in the field, which holds that GBDTs are generally SOTA on tabular data, as opposed to domains such as images and text where DL is the standard. The primary explanation for DL's success in domains such as images and text is that, for these modalities, we have huge amounts of unsupervised data, and have developed architectures and self-supervised pretraining objectives demonstrably capable of learning good internal representations of the data which can be transferred to other tasks with a relatively small amount of supervised fine-tuning data. Despite repeated attempts to replicate this finding in the tabular domain, GBDTs remain state-of-the-art.

The authors of this paper present empirical results suggesting that the primary reason for this perceived limitation is that the relative sparsity of labeled data in the standard academic benchmarks is not representative of realistic industry datasets. As such, they explore the performance of pretrained DL models which are then fine-tuned on downsampled labeled data. The authors compare this with both GBDTs and non-pretrained DL models, and find that, as in other modalities, DL models can exploit pretraining and even surpass GBDTs when the sparsity is low, as is often the case for industry datasets in practice.

**Audience:**

Yes

**Broader Impact Concerns:**

This work is largely empirical in nature, demonstrating efficacy of deep learning techniques for tabular learning-to-rank tasks. I find no broader impact concerns for this work.

**Claims And Evidence:**

Yes

**Requested Changes:**

1. Since you use different forms of augmentation for SimCLR-Rank and SimSiam, it's unclear whether the differences observed are due to the loss or the augmentation strategy. I realize these augmentation strategies were the ones which were chosen because they performed best for each loss, respectively, however it seems to me that it would be better to use the same augmentation strategy when comparing embeddings qualitatively using tSNE, for example, to rule out the possibility that what you are observing is due to the augmentation strategy as opposed to the loss. This is particularly the case when, based on the results in Table 14 and 15, the alternative augmentation was still quite reasonable.
2. When combining models you finetuned a linear layer over concatenated embeddings from each model. Is there any reason not to simply train this whole combined model end-to-end using the supervised data? (I realize each model was already separately fine-tuned, but perhaps this could even be avoided, and replaced by a single round of fine-tuning the concatenated model directly.)
3. Is there a reason the GBDT and non-pretrained models were not trained using data augmentation for the large-scale real-world dataset?
4. Is there any reason not to try GBDT on the learned embeddings from either the SimCLR-Rank, SimSiam, or concatenated version?

**Strengths And Weaknesses:**

### Strengths
1. **Comprehensive empirical evaluation:** good selection of baselines (including both GBDTs with a variety of data augmentation approaches as well as supervised DL models), with reasonable exploration of hyperparameters and loss objectives.
2. **Demonstrates potential of pretraining for tabular LTR in settings with sparse labels:** the empirical results clearly support the main claim, that pretraining DL models can be leveraged for better tabular LTR models in settings with sparse labels.
3. **Observation of more efficient contrastive learning objective (SimCLR-Rank):** while not extremely methodologically novel, the authors do note that in the LTR setting the available targets to rank for a given query (what the authors call a "query group") provide a natural source of hard negatives, and as such the contrastive learning objective in SimCLR can be made more efficient by summing over only those available targets.
4. **Clear writing:** The authors present the motivation for their approach clearly, and the technical details are also presented well.

### Weaknesses
1. **Introduction and focus on SimCLR-Rank as a novel contribution slightly weakens main storyline:** As described in my summary, my interpretation of the main contribution of this paper is the observation that pretraining for tabular LTR is beneficial in a practical label-scarce setting. On the other hand, my interpretation of SimCLR-Rank is that it is a fairly straightforward adaptation of the existing SimCLR loss which primarily makes it more efficient. This is supported by the fact that SimCLR obtains similar results in Table 1. The observation that ranking tasks generally provide a natural source of "hard negatives" is worth mentioning, but beyond this it seems more like an implementation detail which makes it possible to run on larger datasets then I think the authors could de-emphasize it as a novel contribution.
2. **Discussion of binary user feedback in main paper is too brief:**  While the details were available in the appendix, the writing in the paper itself (Section 4.1.2) could be expanded slightly to give the reader a better understanding of how the binary labels were generated. In particular, it might be useful to simply mention that individual elements from the query groups are sampled according to some stochastic process, and the "percentage of labeled query groups" means "percentage of query groups with at least one label".
3. **Ablation is somewhat limited:** While I found most of the paper very well written, section 4.3 was a bit weaker, both in coverage and depth of the experiments and the extent to which the experiments supported the claims therein. (More comments on this in the "Requested Changes" section.)

---

> ### Author Response · Authors · 2024-09-09
> **Thank you for helpful comments!**
>
> We thank the reviewer for their helpful suggestions and close reading of the paper.
>
> > Since you use different forms of augmentation for SimCLR-Rank and SimSiam, it's unclear whether the differences observed are due to the loss or the augmentation strategy. It would be better to use the same augmentation strategy when comparing embeddings qualitatively using tSNE, for example, to rule out the possibility that what you are observing is due to the augmentation strategy as opposed to the loss.
>
> Thank you for your suggestion. We have updated Figure 4 to show t-SNE plots while keeping the augmentation strategy fixed. We did not see a significant change in results, but we agree that methodologically, doing so removes a possible confounding variable.
>
> > When combining models you finetuned a linear layer over concatenated embeddings from each model. Is there any reason not to simply train this whole combined model end-to-end using the supervised data?
>
> We tried this approach in our preliminary experiments and found empirically that it did not work well. As such, we did not pursue this direction further.
>
> > Is there a reason the GBDT and non-pretrained models were not trained using data augmentation for the large-scale real-world dataset?
>
> We do in fact use the data augmentation strategy of input dropout in non-pretrained DL models for large-scale dataset (see Appendix A.2.5). This strategy is similar to zeroing augmentation strategy during pretraining. We did not use any data augmentation strategy in GBDTs because these boosted ensemble models are not known to require data augmentation for achieving good performance.
>
> > Is there any reason not to try GBDT on the learned embeddings from either the SimCLR-Rank, SimSiam, or concatenated version?
>
> Thanks for the suggestion. We tried this idea and found that while training a GBDT over the learned embeddings helps the performance of the GBDT, it does outperform pretraining followed by deep learners. These results have been added to Table 1, and are reproduced here for your convenience.
>
> | **Method**               | **MSLR (↑)**        | **Yahoo Set1 (↑)**      | **Istella (↑)**        |
> |--------------------------|---------------------|-------------------------|------------------------|
> | Supervised GBDT           | 0.2801 ± 0.0002     | 0.5083 ± 0.0197          | 0.5450 ± 0.0000        |
> | Semi-supervised GBDT      | 0.2839 ± 0.0004     | 0.5061 ± 0.0267          | 0.4656 ± 0.0310        |
> | SimCLR-Rank + GBDT        | 0.3165 ± 0.0050     | 0.5504 ± 0.0135          | 0.5397 ± 0.0098        |
> | SimSiam + GBDT            | 0.3158 ± 0.0030     | 0.5620 ± 0.0167          | 0.5297 ± 0.0051 |
> | SimCLR-Rank               | **0.3868 ± 0.0026** | 0.5843 ± 0.0062          | 0.5609 ± 0.0040        |
> | SimSiam (Chen & He, 2021) | 0.3790 ± 0.0028     | **0.6100 ± 0.0072**      | 0.5189 ± 0.0096        |
>
> > Discussion of binary user feedback in main paper is too brief: While the details were available in the appendix, the writing in the paper itself (Section 4.1.2) could be expanded slightly to give the reader a better understanding of how the binary labels were generated. In particular, it might be useful to simply mention that individual elements from the query groups are sampled according to some stochastic process, and the "percentage of labeled query groups" means "percentage of query groups with at least one label".
>
> Thank you for your suggestion. We updated the Sec 4.1.2 accordingly.

---

> > ### Author Response · Authors · 2024-09-20
> >
> > Dear Reviewer fFBs,
> >
> > Thank you again for your thorough review and valuable feedback. We just wanted to check if you had any additional questions or concerns that we can address. Thank you for your consideration!
> >
> > Best regards,
> > The Authors

---

### Decision · Action_Editor_z4DP · 2024-10-12

**Recommendation:** Accept as is

**Comment:**

I congratulate the reviews and the authors for the discussion. First, it centred on identifying the contribution. The discussion added some empirical details, increasing the paper's scholarship. The changes in the manuscript made the paper more clear.

**Audience:**

Learning to rank in the presence of tabular data is an important problem in machine learning. The paper is of interest both for that problem, and for the applicability of deep learning methods in settings where it hasn't succeeded before.

**Claims And Evidence:**

The paper consider the problem of learning-to-rank, studying the question of whether deep learning methods can outperform Gradient Boosted Decision Trees (GBDTs) in tabular data. Under label-scarcity, the paper shows that deep learning methods can outperform GBDTs.
The paper revise the literature, and perform extensive experiments in the propose methods.

The reviewer have a positive initial reaction to the paper. The discussion improved certain aspects, making it more clear how it fits in the literature. I coincide with one of the reviewers who point out that the unsupervised pre-training method might have impact beyond this specific setting.